# Research on the Process, Energy Consumption and Carbon Emissions of Different Magnesium Refining Processes

**DOI:** 10.3390/ma16093340

**Published:** 2023-04-24

**Authors:** Jingzhong Xu, Tingan Zhang, Xiaolong Li

**Affiliations:** School of Materials and Metallurgy, Northeastern University, Shenyang 110819, China

**Keywords:** silico-thermic, aluminothermy, carbothermic, energy consumption, carbon emission

## Abstract

Under the policy of low carbon energy saving, higher requirements are put forward for magnesium smelting. As the mainstream magnesium smelting process, the Pidgeon process has the disadvantages of a long production cycle, high energy consumption and high carbon emission, which makes it difficult to meet the requirements of green environmental protection. This paper reviews the research progress on different magnesium smelting processes and further analyzes their energy consumption and carbon emissions. It is concluded that the standard coal required for the production of tons of magnesium using the relative vacuum continuous magnesium refining process is reduced by more than 1.5 t, the carbon emission is reduced by more than 10 t and the reduction cycle is shortened by more than 9.5 h. The process has the advantages of being clean, efficient and low-carbon, which provides a new way for the development of the magnesium industry.

## 1. Introduction

Magnesium is widely used in the aerospace, aviation, automobile, metallurgy and chemical industries due to its low density, high strength, specific rigidity and good die-casting performance [1,2,3]. It has good electromagnetic shielding and shock absorption, making it useful in the field of electronic technology where precision equipment also plays an extremely important role, known as the ‘21st century the most development and application potential of green engineering materials’. Magnesium resources are abundant, mainly in the form of solid magnesium-containing minerals and magnesium-containing evaporative minerals (seawater, brine, salt lake) [4,5,6,7,8]. At present, the reserves of magnesium-containing resources meet human needs. Natural brine can be regarded as a recyclable resource, and mined magnesium will be regenerated in a short time. China has rich resource reserves, energy advantages and smelting policies. Since 1998, China’s magnesium production has ranked first in the world, and magnesium ingots, magnesium alloys and other magnesium products have been exported worldwide. The output of magnesium metal in China and the world from 2014 to 2021 is shown in Figure 1 [9]. China’s production of primary magnesium ingots fell sharply in 2018 due to the novel coronavirus pandemic, but it remains the world’s largest supplier of magnesium ingots, accounting for more than 80% of global production. The Pidgeon process has become the main magnesium smelting process in the world because of its fast construction, small investment and diversified heat sources [10], but it has the disadvantages of a long production cycle, high energy consumption and large emissions. With the continuous research and improvement in magnesium smelting technology at home and abroad, the mechanization and automation level of smelting technology and working conditions have been improved, the production energy consumption has been reduced and some waste slag has been recycled. However, its intermittent production characteristics seriously restrict the production of magnesium, and there is a cooling process between the pellet calcination and reduction process, resulting in a large amount of energy loss. Many researchers have studied the influence on the dynamic performance, heat transfer performance and processing parameters of the magnesium smelting process. However, for the smelting process, energy consumption and carbon emissions are equally important. With the increase in climate temperature year by year, low-carbon emission reduction has become a hot topic in current society. In order to ensure a healthy and comfortable living environment and improve people’s quality of life, the smelting method for metal magnesium must have a good economy, low-carbon emission, production continuity, excellent product quality, reliable production stability and production safety. Based on the study of energy consumption and carbon emission of different magnesium smelting methods, this paper compares and analyzes the electrolysis method, silico-thermal reduction method, carbon thermal reduction method and aluminum thermal reduction method. The energy consumption and environmental impact are further reviewed to provide a reference for scientific research in the field of magnesium smelting.

According to statistics from the Magnesium Industry Branch of the Non-ferrous Metals Association [11], the demand for raw magnesium is concentrated in the metallurgical field (aluminum alloy addition, steelmaking desulfurization, ball milling cast iron and metal reduction) and processing field (rare earth magnesium alloy, castings, die castings and profiles). The domestic original magnesium capacity concentration is low, with more than 30,000 t of production capacity from only 13 companies for a total market share of 35.9%. The distribution of domestic magnesium smelters and raw magnesium production in 2020 is shown in Figure 2 [12,13,14,15,16,17,18,19].

The magnesium refining methods are divided into thermal reduction and electrolysis, and the sources of raw materials are magnesite, dolomite, hydromagnesite, halloysite, serpentine and seawater [15,16,17,18,19]. These raw materials differ in magnesium content, production methods and sources, and the main source methods are mining, open pit mining, seawater and salt lake reprocessing, asbestos production waste, etc. Magnesium is obtained from different magnesium sources, usually as natural raw materials, and rarely in pure form. The main properties of different raw materials are shown in Table 1. The raw materials for magnesium smelting are mainly magnesite and serpentine. The distribution of magnesite production is shown in Figure 3a, and the distribution of serpentine production is shown in Figure 3b.

**Table 1 materials-16-03340-t001:** Main properties of raw materials for magnesium smelting [10,11,12,13,14,15,16,17,18,19,20,21,22,23,24,25,26,27,28,29,30,31,32,33,34,35,36,37,38,39,40,41,42].

Raw Material	Essential Component	ORE Color	Crystalline Phase	Hardness	Density (kg/m^3^)	Reserve (Million Tons)
Magnesite	MgCO_3_ [10]	White [10]	Tripartite crystal system [20]	4–4.5 [12,20]	2.9–3.1 [13]	40 [14]
Dolomite	CaMg(CO_3_)_2_ [21]	Off-white	Tripartite crystal system [22,23]	3.5–4 [24]	2.8–2.9 [25]	85 [26]
Bichofite	MgCl_2_·6H_2_O [27]	White	Monoclinic system [28]	1–2	——	17.74% of salt lake brine [29]
Carnallite	KCl·MgCl_2_·6H_2_O	White–red	Hexahedron [30]	——	1.6	36.8% of the Dead Sea [31,32]
Serpentine	Mg_6_[Si_4_O_10_](OH)_8_	Waxy luster	Monoclinic system [33,34,35]	2.5–4 [36]	2.57 [37]	120 [38]
Seawater	Table 2 [39,40,41]	Dark blue	——	——	1.02–1.07 [42]	——

## 2. Methods for Magnesium Smelting

The mature magnesium smelting methods in the world’s magnesium smelting industry are divided into two categories. One is the molten salt electrolysis method. In the molten electrolyte of magnesium chloride, the metal magnesium is directly obtained with direct current electrolysis. Based on the different raw materials used, it can be divided into three methods for smelting magnesium metal with magnesite, brine and carnallite as raw materials. Another is the thermal reduction method, which uses calcined dolomite as the raw material under high-temperature conditions to reduce magnesium oxide into magnesium metal. Depending on the reducing agent, it is divided into the silico-thermal method, aluminum thermal method and carbon thermal method. The worldwide use of electrolytic magnesium manufacturers is high, and its output once accounted for 80% of the world’s total magnesium production. In recent years, with the increasing production of magnesium in China, the proportion of metal magnesium produced with the silico-thermal method has increased significantly. In 2007, China produced 67 million tons of primary magnesium, accounting for 85% of the world’s total production and 75% of the world’s total production of magnesium metal produced with the silico-thermic process. Electrolytic magnesium smelting has problems related to a complex production process, high power consumption, poor production conditions and serious equipment corrosion. The leakage of chlorine gas seriously pollutes the environment and affects people’s health. The waste gas, wastewater and waste residue are large, and the treatment cost is high. The problems of expensive reducing agents, the short life of the reduction tank, the low capacity of a single tank and low production efficiency exist in the silico-thermic process for magnesium smelting. The Zhang Ting-an team at Northeastern University proposed a new concept of the ‘Relative vacuum’, breaking through the thermodynamic limitations of the original technology that must be produced in a vacuum environment, and constructing a new theory, process and equipment for the ‘relative vacuum continuous magnesium smelting’, which provides a theoretical basis for the continuous smelting of metal magnesium. The rapid and continuous smelting of metal magnesium under actual micro-positive pressure conditions was realized, and the reduction cycle shortened from 10~14 h when using the Pidgeon method to 1~1.5 h, with a comprehensive recovery rate for magnesium of more than 90%. To achieve a carbon emission reduction per ton of magnesium metal consumption of standard coal from 5 t to 3~3.5 t and CO_2_ emission reduction from 23 t to 11~13 t, the solution for the current magnesium smelting method cannot be continuous production, labor insensitive, heavy pollution or low production efficiency. The industrial induction vertical-tank device for magnesium smelting is innovatively designed, which greatly improves the service life of the reduction tank and solves the problems of a slow heat transfer speed, long heating time and short service life of the reduction tan. It also lays an industrial foundation for the continuous production of magnesium metal.

### 2.1. Silico-Thermal Method

The silico-thermal method has become the mainstream method for magnesium smelting because of its advantages including fast construction, small investment, wide heat source and high product quality. However, there are some problems such as large fuel consumption, low productivity, a low energy utilization rate and serious air pollution. Researchers at home and abroad have focused on the optimization of silico-thermal process parameters, kinetic analysis, reactor model establishment, industrial equipment, low carbon, energy savings and emission reduction. This work has achieved good results.

Li Rongbin et al. [43] studied the reduction process of a single pellet in the reactor and concluded that the reduction degree of magnesium increased with the decrease in pellet thickness and the increase in temperature, and the reduction efficiency was highest at 1373~1473 K. The kinetic analysis showed that the reduction reaction of magnesium is affected by the phase interface model and the diffusion model. When the reduction temperature is increased by 50 °C, the reduction degree and reduction rate of magnesium can be increased by 5~10%, and the production cycle is shortened accordingly. Surendran. Ghosh et al. [44] studied the form of dicalcium silicate and concluded that the γ-type is stable at room temperature without any stabilizer, and the β-type exists in silicate. The hydraulic performance of γ-type dicalcium silicate is low, but it shows quite high hydraulic performance in the case of excessive CaO. α-dicalcium silicate has excellent hydraulic performance under certain stabilizers, and the kinetic performance of β-C_2_S is similar to that of C_3_S. L.M. Pidgeon [45] measured the equilibrium pressure of magnesium vapor in the system 2MgO + 2CaO + Si(Fe) between 1100 °C and 1200 °C using the entrainment method and further calculated Kp and ∆H using P_Mg_. It was concluded that CaF_2_ does not affect the equilibrium vapor pressure and is an effective catalyst for the reaction. This study further verified that the solid diffusion between reactants affects the reaction rate. Hangjin Wu et al. [46] prepared magnesium metal using dolomite and ferrosilicon in an electrothermal coupled field at 10 Pa. The effects of reaction time, DC electric field strength and current density on the reduction rate were investigated, and the reduction mechanism was explained with the Joule heat effect. It was concluded that the reduction rate of magnesium oxide increased significantly with the increase in reaction time, the decrease in DC field strength and the increase in current density. The reduction rate reached 88.35% with a heating time of 150 min under the conditions including a DC field strength of 950 V/cm, device temperature of 700 °C and current density of 1.18 A/cm^2^. The purity of the generated magnesium metal was 98.54 wt%. Chao Wang et al. [47] investigated the silicon-thermal reduction of the magnesium refining process by varying the experimental temperature, CaF_2_ content and reaction time. Under the conditions of experimental temperature from 1323 K to 1473 K and CaF_2_ content from 0% to 3%, the required reaction time was correspondingly shortened, and the reaction rate was correspondingly accelerated with the increase in CaF_2_ content. Further Box–Lucas fitting with temperature, CaF_2_ content and reaction time as independent variables was carried out for the silico-thermal reduction process of magnesium, and it was concluded that CaF_2_ affects the diffusion process rather than the reduction process. Majid Maarefvand et al. [48] studied the failure principle of distillers using thermodynamic simulation and microstructural analysis. It was concluded that grain boundary sliding causes cracks to form at grain boundaries, microcracks connecting grain interiors cause cracks to form inside the grain and austenite decomposes into pearlite in specific regions.

Jibiao Han et al. [49,50] studied the nucleation and condensation mechanism of Mg vapor carried by argon and concluded that the condensation temperature of Mg vapor is influenced by the heat source temperature, the partial pressure of Mg vapor and the flow rate of argon. When the heat source temperature was 1473 K and the argon flow rate was 0.2 m^3^/h, the condensation temperature of magnesium vapor was 1013.3 K. The increase in Mg vapor supersaturation and the decrease in condensation temperature were favorable to the growth of liquid nucleation. The higher the purity of the condensation product, the less likely it was to be oxidized in the argon gas stream. Using a further experiment, it was concluded that when the partial pressure of magnesium vapor is lower than 371.2 Pa, the condensation phenomenon is produced; however, when higher than 371.2 Pa, the liquefaction process of magnesium vapor exists. Yusi Che et al. [51] optimized the design of a thermogravimetric analyzer, the schematic diagram of which is shown in Figure 4. The silico-thermal reduction CaO·MgO reaction under vacuum and high-temperature conditions was investigated. The thermogravimetric losses in the pellets were determined by varying the heating rate (1.5, 2.0, 2.5, 3.0 °C/min) at 5 Pa vacuum and 300 to 1400 °C. The thermogravimetric curves for the starting and ending sections of the reaction were analyzed in terms of shape characteristics, the half-width ratio of differential thermogravimetry and kinetic parameters, yielding an activation energy of 233.42 KJ/mol and a finger-front factor of 5.14 × 10^10^ s^−1^.

Carol L. Steen et al. [52] studied the kinetics and structure of the hemicalcification reaction and concluded that at 710~800 °C, the rate of hemicalcification increases with increasing temperature and decreasing grain size independent of the external air pressure and CO_2_ pressure. The reaction rate is similar to the pearlite reaction and is controlled by the growth rate of hemicalcified dolomite at the active growth sites at the grain boundaries or edges. S. K. Barua et al. [53] introduced magnesium vapor into hydrogen and studied the reaction kinetics for the transfer of the reaction to the products at temperatures ranging from 1070 to 1250 °C. It was determined that the overall rate of the reaction under a hydrogen environment was higher than that under vacuum conditions. It was further concluded that the effect of the diffusion process on the overall reaction rate increases with decreasing pellet porosity and increasing particle size. I.M. Morsi et al. [54] studied the kinetics of the silico-thermal reduction process under inert gas protection. It was concluded that the molar ratio of calcium oxide to magnesium oxide influences the products of the silico-thermal reduction reaction. The suitable ratio of silicon in ferrosilicon was 1.45. Calcium fluoride with a weight percentage of 2.5 was found to facilitate the reduction process. The pellets were pressed at 450 MPa, and the reduction rate was higher by about 92%. The silicon-thermal reduction process is controlled by the diffusion of reactants in a solid-state reaction with an apparent activation energy equal to 306 KJ/mol. J.R. Wynnyckyj et al. [55] determined the calcium pressure in Ca-Si alloys using the Knudsen cell and the magnesium and calcium pressures on CaO + MgO + Si mixtures using the transport method. The improved value for the standard free energy of calcium orthosilicate at 1400 K was obtained at −33,400 cal. Liu Xiaoxing et al. [56] improved the conversion rate, α, in the kinetic parameter Freeman Carroll method for the thermal decomposition reaction of solids. The activation energy, E, for the thermal decomposition of Mg(OH)_2_ was obtained at 122 KJ/mol, and the number of reaction steps, n, was 0.68. The results obtained using the improved algorithm were more accurate. The improved calculation formula for the conversion rate, α, is shown in Formula (1).
(1)$$\frac{d\alpha }{dT}=-\frac{1}{W_{0}-W_{\infty }}\cdot \frac{dW_{x}}{dT}$$
where *α* is the conversion rate, *W_x_* is the thermal gravimetric rate at any point in the thermogram, *W*_0_ and *W*_∞_ are constants and *T* is the dynamic thermal decomposition temperature.

Chao Zhang et al. [57] investigated the kinetic mechanism for the decomposition and magnesium reduction stages using solid-state kinetic modeling, combining decomposition and reduction in a reactor to establish a one-step model for chemical reaction kinetics and heat transfer in decomposition and reduction. The effect of heating temperature on the degree of decomposition and reduction in the two stages was simulated. The analysis shows that the one-step technology can effectively reduce the cycle time of the dolomite decomposition stage and magnesium reduction stage, and it can collect the high temperature carbon dioxide produced by decomposition in time, while reducing the heat loss and saving energy significantly. The kinetic analysis of silica-thermal reduction of calcined dolomite under argon atmosphere performed by W. Wulandari et al. [58] concluded that solid-phase diffusion of reactants controls the reaction rate. The effect of mass transfer of magnesium vapor from the surface to the main gas phase shows that gas film mass transfer does not affect the kinetic rate and that solid-state diffusion is the main influence. R.B. Li et al. [59] investigated the effect of different constant heating temperatures on the reduction process by developing a model for magnesium reduction with heat transfer and chemical reaction kinetics. It was concluded that the heat transfer of the pellet, the external heat flow of the distillator and the reaction kinetics all affect the magnesium reduction process in the distillator, and the degree of magnesium reduction increases with the increasing heat transfer rate. The reduction reaction of magnesium takes place in the pellet layer, and the reduction process gradually advances from the external region of the distiller to the internal one, and the external heat flow makes the reduction degree asymmetric. The effect of different hydration activities of individual pellets on the chemical reaction kinetics of the silicon-thermal reduction process was first experimentally investigated by Chao Zhang et al. [60]. It was concluded that the hydration activity affects the intrinsic chemical reaction rate of the silicon-thermal reduction process but not the final reduction rate. A numerical model with coupled radiation, heat conduction and chemical kinetics was developed to simulate the heat transfer phenomenon in the silico-thermal reduction process. It was concluded that an increase in hydration activity is beneficial for reducing the reduction cycle time and increasing the production capacity. Yusi Che et al. [61] studied the thermogravimetric weight of pellets made with calcined dolomite, ferrosilicon and fluorite at four heating rates (10, 15, 20 and 25 °C) and performed reaction kinetics using a non-isothermal thermogravimetric analysis technique under argon protection and high temperature. It was concluded that the reaction mechanism is determined by N-dimensional nucleation and growth when the conversion rate, α, is less than 48.6%; otherwise, it is an N-level formal chemical reaction. Based on the kinetic model analysis, the kinetic function for the solid phase reaction is described according to Table 3 under three assumptions. Firstly, the reaction consists of several basic reaction steps, and the conversion rate of each step can be described with its own kinetic equation. The second assumption is that all kinetic parameters, such as E, A, n (reaction order) and f(α), are assumed to be constants for each individual reaction step. The third assumption is that the thermal analysis signal is the sum of the signals in a single reaction step, and the effect of each step is calculated as the conversion rate multiplied by the effect of the step. In the table, α is the conversion rate, f(α) is the differential form of the kinetic function for the solid phase reaction, g(α) is the integral form of the kinetic function for the solid phase reaction, E is the apparent activation energy, LogA is the pre-exponential constant, n is the dimensionless reaction order and R^2^ is the correlation.

M. Morsi et al. [62] studied the kinetics of the silica-thermal reduction of dolomite by varying the sintering temperature, reduction temperature and reduction time in a magnetothermal reactor protected with argon gas. A schematic diagram of the magnetothermal reactor is shown in Figure 5. It was found that the reduction rate can reach 80% at the sintering temperature of 700 °C for 1 h and the reduction temperature of 1550 °C for 15 min. The reduction of dolomite is a solid-state diffusion process with an activation energy of about 498 KJ/mol. In the silicon-thermal reduction reaction, the mixture of calcium oxide and silicon forms a liquid Ca-Si alloy at a temperature of 1000 °C.

Huaqiang Chu et al. [63] established a single homogeneous chemical reaction model under the condition of silico-thermal reduction and studied the kinetic principle of the reduction process. The equation for the 1273~1473 K chemical reaction model is shown in Formula (2). It was concluded that low heat transfer efficiency is an important factor limiting magnesium production capacity in the initial reduction stage. The radiation intensity changes with temperature, and the radiation heat transfer affects the heat transfer process and cannot be ignored or simplified.
(2)1−1−α13=k0−ERTτ
where *α* is the absorption coefficient, *k*_0_ is the pre-constant (min^−1^), *T* is the local temperature (K) and *τ* is the reduction time of magnesium in the experimental and numerical models (min).

Zhang et al. [64] studied the reaction and heat transfer mechanism of magnesium smelting using the silico-thermal method by establishing an unsteady model for radiation and heat conduction coupling. It was concluded that the minimum temperature of the pellets is 1203 K after 2 h of production, and the average reduction rate of the materials is 66% after 4 h. With the extension of calcination time, the reaction area was close to the center of the reduction tank. Due to the decrease in pellet quantity and heat transfer efficiency in the central area, the magnesium yield in the first half cycle was much higher than that in the second half cycle. The pellets are arranged in the reduction tank as shown in Figure 6.

Majid Shahheidari et al. [65] reduced magnesium production costs by adding brackets in the middle of the vulnerable tube and changing the cross-section to extend the life of the tank. The study found that the service life was significantly improved after adding the bracket. Using economic analysis, it was determined that the optimal design for the magnesium production steel tank is an external reinforcement bracket that is 50 cm long and 10 mm thick. A comparison between the supported vulnerable pipe and the original pipe before and after use is shown in Figure 7.

Wen Ming et al. [66] proposed a new process for magnesium smelting using the silico-thermal method, which is to make pellets before calcination. In the low temperature stage of 1073 K held for 30 min, the burning loss rate of MgCO_3_ decomposition was 18.43%, and in the high temperature stage of 1273 K held for 30 min, the burning loss rate of CaCO_3_ decomposition was 21.06%. It was concluded that staged calcination can shorten the calcination decomposition process. J.M. Toguri et al. [67] found that Mg_2_SiO_4_ and Si (brownish yellow, glassy deposition) outside the reaction zone caused a large weight loss in the reactants. It was proposed that silicon and silicate are transferred in the form of gaseous SiO. The reaction equation can be expressed as:(3)$$\left(4n+1\right){\mathrm{MgO}}_{\left(s\right)}+\left(1+n\right){\mathrm{Si}}_{\left(s\right)}={\mathrm{nMg}}_{2}{\mathrm{SiO}}_{4}{}_{\left(s\right)}+\left(1+2n\right){\mathrm{Mg}}_{\left(g\right)}+{\mathrm{SiO}}_{\left(g\right)}$$

Hongzhou Ma et al. [68] added aluminum to a ferrosilicon reductant to reduce the reaction temperature. It was concluded that at a temperature of 720 °C and a vacuum pressure of 10 Pa, aluminum reacts with magnesium oxide to form magnesium vapor and alumina, and alumina reacts with calcium oxide to form calcium aluminate. When the temperature rises to 1150 °C, silicon begins to reduce magnesium oxide to form silicon oxide and then reacts with calcium oxide to form calcium silicate. When the temperature continues to rise, both aluminum and silicon will participate in the reduction of magnesium oxide. Mehmet Bugdayci et al. [69] studied the effect of the type and amount of reducing agents on the reaction. It was concluded that CaC_2_ slightly reduces the recovery of magnesium but significantly reduces the process cost. At 1250 °C for 360 min, the recovery of magnesium was 94.7% when 20% CaC_2_ was added. Under the condition of 1200 °C for 300 min when using aluminum as a reducing agent and reducing the reaction temperature, the recovery rate of magnesium was 88.0%. Yusi Che et al. [70] proposed a new magnesium production technology using a thermogravimetric analyzer (TGA) to carry out a thermogravimetric and derivative thermogravimetric analysis of pellets at different heating rates. When the argon temperature was higher than 1343 K and the inlet velocity was greater than 0.05 m·s^−1^, the cycle time of the new technology could be reduced. The effect of argon temperature on the reduction degree was much greater than that of inlet velocity. Yang Tian et al. [71] compared and analyzed the Pidgeon method and the vacuum carbothermal reduction method and concluded that the vacuum carbothermal reduction process has low energy consumption (2.675 tce:8.681 tce), a low material magnesium ratio (2.953:1 vs. 6.429:1), low carbon emissions (8.777 t:26.337 t), and low solid waste production (0.522 t:5.465 t). The vacuum carbothermal reduction process consumed 5.656 t of non-renewable mineral resources, which was 9.689 t less than that of the Pidgeon process. Yifei Wang et al. [72] used Spent Pot Lining First Cut (SPL-1cut) as an alternative fuel for dolomite calcination in the Pidgeon process. At 850 °C for 120 min, the cyanide was completely oxidized and decomposed by adding 40% dolomite to the SPL-1cut. The dolomite was decomposed into MgO and active CaCO_3_. NaF reacted with active CaCO_3_ and converted to CaF_2_. Rongbin Li et al. [73] compared the magnesium reduction rate and production efficiency of the short process and the Pidgeon process and studied the effects of thermal decomposition temperature, silico-thermal reduction temperature, granulation pressure and the silicon ratio on the production of magnesium in the short process. They also analyzed the cross-linking effect between the carbon dioxide decomposed from dolomite and the metal elements in ferrosilicon. It was concluded that the effect of decomposition and reduction temperature on the reduction process of magnesium is more important than the granulation pressure and silicon ratio. A decomposition temperature of <1000 °C and a reduction temperature of >1100 °C can improve the magnesium reduction rate and production efficiency of the new short process.

### 2.2. Aluminothermic Method

At present, the international mainstream magnesium refining method is still the Pidgeon silico-thermal method for magnesium refining, but its high reaction temperature (1200 °C), large material to magnesium ratio (production of 1 t magnesium consumption 10 t dolomite), short working life of the reaction tank and other factors seriously limit its development. Under the policy of low carbon emission reduction, a difficult task is put forward for the magnesium refining industry. Aluminothermic reduction magnesium refining technology has the advantages of low raw material consumption, high production efficiency, low energy consumption, no waste slag emission, etc., and has good application prospects. Researchers at home and abroad have focused on the optimization of aluminothermal process parameters, kinetic analysis, reactor modeling, energy saving and emission reduction. This work has achieved good results. The silico-thermal reduction reaction equation is shown in Equation (4), and the aluminum thermal reduction reaction equation is shown in Equations (5)–(9) [74].
(4)$${2\mathrm{MgO}}_{\left(s\right)}{+2\mathrm{CaO}}_{\left(s\right)}{+\mathrm{Si}}_{\left(s\right)}{=2\mathrm{Mg}}_{\left(g\right)}{+\mathrm{Ca}}_{2}{\mathrm{SiO}}_{\left(s\right)}$$
(5)$$12\mathrm{Ca}O_{\left(s\right)}+21MgO_{\left(s\right)}+14Al_{\left(l\right)}=21Mg_{\left(g\right)}+12CaO\cdot 7Al_{2}O_{3}{}_{\left(s\right)}$$
(6)$${\;\mathrm{CaO}}_{\left(s\right)}{+3\mathrm{MgO}}_{\left(s\right)}{+2\mathrm{Al}}_{\left(l\right)}{=3\mathrm{Mg}}_{\left(g\right)}+\mathrm{CaO}\cdot {\mathrm{Al}}_{2}O_{3}{}_{\left(s\right)}$$
(7)$${\mathrm{CaO}}_{\left(s\right)}{+6\mathrm{MgO}}_{\left(s\right)}{+4\mathrm{Al}}_{\left(l\right)}{=6\mathrm{Mg}}_{\left(g\right)}+\mathrm{CaO}\cdot {2\mathrm{Al}}_{2}O_{3}{}_{\left(s\right)}$$
(8)$${4\mathrm{MgO}}_{\left(s\right)}{+2\mathrm{Al}}_{\left(l\right)}{=3\mathrm{Mg}}_{\left(g\right)}+\mathrm{MgO}\cdot {\mathrm{Al}}_{2}O_{3}{}_{\left(s\right)}$$
(9)$${3\mathrm{MgO}}_{\left(s\right)}{+2\mathrm{Al}}_{\left(l\right)}{=3\mathrm{Mg}}_{\left(g\right)}{+\mathrm{Al}}_{2}O_{3}{}_{\left(s\right)}$$

Teng Zhang et al. [75] subjected a mixture of MgO:Al:Ca(OH)_2_ with a molar ratio of 3:2:1 to aluminothermic reduction at atmospheric pressure and temperature of 1223 K. It was concluded that high-energy ball milling can strengthen the aluminothermic reduction mixture of magnesium oxide and calcium hydroxide and successfully prepare high purity magnesium crystals. The particle size of magnesium powder increased with the increase in ball milling time and decreased with the increase in sodium stearate content. It was also concluded that high-energy ball milling led to a decrease in the apparent activation energy of the reaction. The solid phase reduction deposit phase mainly consisted of 12CaO·7Al_2_O_3_ and Al_2_O_3_. The MgO-CaO-CaF_2_ was reacted with Al-C powder to extract magnesium under atmospheric pressure and 1223 K using mechanical activation technology [76]. It was concluded that the reduction rate of magnesium oxide is improved with the increase in C, CaO and CaF_2_ content and the prolongation in ball milling time. Adding C powder to the reactants during ball milling can enhance the reactivity of Al. When the CaO content is 20~30 wt%, the reaction process is MgO reduction to form 12CaO·7Al_2_O_3_, magnesium aluminate and CaAl_1.9_O_4_C_0.4_, whereas when the CaO content is more than 30 wt%, magnesium aluminate phase disappears. With the addition of CaF_2_, the reduction reaction is characterized by the reduction of MgO, generating 11CaO·7Al_2_O_3_·CaF_2_ and CaAl_1.9_O_4_C_0.4_. The reaction mechanism is shown in Figure 8.

Yaowu Wang et al. [77] studied the vacuum aluminothermic reduction of dolomite and magnesite. The slag was mainly A1_2_O_3_ (more than 67%, in the form of CaO·2A1_2_O_3_) and CaO. The process flow is shown in Figure 9. When the temperature was 95 °C with the liquid–solid ratio of 5 and the leaching time of 2 h, the mixture of NaOH and Na_2_CO_3_ is used to leach aluminum hydroxide from the slag, and the leaching rate of alumina is greater than 86%. The two main reasons for the loss of alumina in the reduction process are the formation of magnesium aluminate spinel and the leaching of 3CaO·Al_2_O_3_·6H_2_O. The chemical composition of aluminum hydroxide obtained after carbonation precipitation can meet the quality standards of an alumina plant.

Lan Hong [78] studied the reaction mechanism of the aluminothermic reduction process in the temperature range of 1273~1873 K. In the reaction process, magnesium oxide is first reduced by aluminum to form Mg_(g)_ and aluminum magnesium spinel (MgO·Al_2_O_3_), and the reduction degree is affected by the pellet-forming pressure. When the molding pressure is high, the contact area of the sample increases, which is conducive to the reduction reaction. The generated spinel is then reduced by aluminum to slowly generate magnesium vapor. At different heating rates, the activation energy of the second stage weight loss reaction was determined to be 151.2 KJ/mol. It was further concluded that the addition of lime to the pellets can accelerate the reaction because the melting point of the compound of alumina and lime is lower than that of spinel.

Peng Deng et al. [79] studied the effects of different molding pressure, reaction times and material ratios on the aluminothermic reduction process. It was concluded that the increase in forming pressure can reduce the porosity of pellets, increase the heat transfer coefficient and increase the contact area of the aluminothermic reaction. With the increase in reaction time and the decrease in the Mg/Al molar ratio, the reduction rate of magnesium oxide increases. The reduction rate of magnesium oxide can reach 91.33% under the conditions of a magnesium–aluminum molar ratio of 3:2, molding pressure of 200 MPa, reaction temperature of 1200 °C and reaction time of 5 h. The aluminothermic reduction reaction is first carried out in the outer layer of the pellets, and the aluminum element migrates from the inside to the outside, which makes the phase composition of the pellets different. The aluminum content in the outer layer of the pellet is high, and the phase is mainly alumina and magnesium aluminum oxide. The aluminum content in the inner layer of pellets is low, and the phase is mainly magnesium aluminate spinel and unreacted magnesium oxide. When the reaction temperature is lower than 1000 °C, the aluminothermic reduction reaction rate is controlled by the interfacial chemical reaction rate under the condition of a Mg:Al molar ratio of 3:2 and molding pressure of 10 MPa. When the temperature is above 1100 °C, the reaction rate is controlled by the diffusion rate through the product layer. Jian Yang et al. [80] placed magnesium oxide powder and aluminum powder into a quartz tube and achieved an isothermal reduction environment by controlling the position of the quartz tube in a graphite crucible. The reduction rate increased with the increase in temperature, carrier gas flow rate and balling pressure, and the carrier gas flow rate had the greatest influence on the reduction rate of magnesium oxide. The aluminothermic reduction of magnesium oxide was divided into two stages. The first produced spinel, alumina and magnesium vapor. The spinel was then reduced by aluminum to produce alumina and magnesium vapor, and the excess magnesium oxide was also reduced. The kinetic model was established, and the apparent activation energy of the reduction reaction was 109 KJ/mol. Jianping Peng et al. [81] added calcium carbonate to magnesium borate for calcination. After calcination, the magnesium element was in the form of MgO. Part of CaO generated by CaCO_3_ after calcination reacted with B_2_O_3_ to form Ca_3_B_2_O_6_, which was then mixed with aluminum powder to prepare magnesium metal. It was found that the aluminothermic reduction of boehmite has a high reduction rate (93%) under the conditions of pellet pressure of 45~60 MPa, reduction temperature of 950~1000 °C and reduction time of 0.5 h. It was concluded that the increase in briquetting pressure is helpful to the full contact between materials and the reduction reaction. However, when the pelletizing pressure is too large, it will cause poor permeability, and the moisture and CO_2_ escape resistance increases, which is not conducive to the reduction reaction. Increasing the calcination temperature or prolonging the calcination time can promote ore decomposition. However, when the calcination temperature is too high or the calcination time is too long, the magnesium reduction rate will be reduced.

Jian Yang et al. [82] studied the behavior of magnesium vapor generated with the aluminothermic reduction of magnesium oxide in the process of hot metal desulfurization. It was found that the curve of magnesium concentration is parabolic. The magnesium concentration in molten iron increased at the initial stage of dissolution, and then decreased with the evaporation on the melt surface and the mass transfer of dissolved magnesium to the bubble surface. When the temperature, pellet mass and carrier gas flow rate increased, the magnesium concentration increased. With the increase in initial sulfur content, the magnesium concentration decreased. DaXue Fu et al. [83] conducted aluminothermic reduction of a mixture of calcined dolomite and calcined magnesite. The reduction process was divided into three stages. The first stage is 0 ≤ ηt/ηf ≤ 0.43 ± 0.06. The direct reaction of calcined dolomite or calcined magnesite with Al produces 12CaO·7Al_2_O_3_ and magnesium aluminate spinel. The process is controlled by the penetration of aluminum liquid into the magnesium oxide phase and the chemical reaction between aluminum and magnesium oxide. The second stage is 0.43 ± 0.06 ≤ ηt/ηf ≤ 0.9 ± 0.02, and the product is CaO·Al_2_O_3_. The diffusion and chemical reaction of Ca^2+^ and aluminum melt affect the total reaction rate. The third stage is 0.9 ± 0.02 ≤ ηt/ηf < 1, and the product is CaO·2Al_2_O_3_, and the reaction process is controlled by Ca^2+^ diffusion. Here, ηt and ηf are the reduction rate at time t and the final reduction rate obtained at temperature T in the experiment, respectively. Feng Gao et al. [84] studied the thermodynamic parameters of the reduction of magnesium with different reducing agents including ferrosilicon, aluminum, silicon and the aluminothermic method. It was found that the initial reaction temperature of aluminum was 906.38 K lower than that of silicon for the direct reduction of MgO at standard atmospheric pressure with only aluminum or silicon added. When CaO was added to the raw material, the initial reaction temperature of aluminum was 755.63 K higher than that of silicon. The increase in temperature in the reaction zone increases the equilibrium vapor pressure of magnesium, increases the difference between the actual pressure of magnesium vapor in the reaction zone and the equilibrium pressure, and accelerates the reduction reaction rate. Because of the low melting point, the reaction rate with Al as the reducing agent is faster than that with Si as the reducing agent. Increasing the vacuum degree can reduce the residual pressure in the reaction system, which is beneficial to the diffusion of magnesium vapor from the reaction zone and can accelerate the reduction reaction speed, prolong the average free path of magnesium atom diffusion and obtain crystal dense magnesium. The increase in vacuum degree is beneficial to the utilization rate of the silicon reducing agent. XianXi Wu et al. [85] studied the reaction equilibrium magnesium vapor pressure under different reduction methods, as shown in Formulas (10)~(12). *LgP^°^_Mg_* represents the magnesium vapor pressure of the reaction between aluminum and calcined dolomite. *LgP_Mg_* represents an experimental measure of the magnesium vapor pressure of the reaction between Al-Si alloy and calcined dolomite. *LgP^∆^_Mg_* represents the magnesium vapor pressure measured with Pidgeon method. It was concluded that the reaction temperature is 1000 °C when aluminum and high aluminum silicon alloy replace ferrosilicon in the reduction reaction.
(10)$$LgP_{Mg}^{{}^{\circ}}=-\frac{8186}{T}+7.982(900\sim 1050{}^{\circ}C)$$
(11)$$LgP_{Mg}=-\frac{8853}{T}+8.455(900\sim 1056{}^{\circ}C)$$
(12)$$LgP^{\Delta }{}_{Mg}=-\frac{10875}{T}+8.918(1100\sim 1200{}^{\circ}C)$$

Xing Xie [86] established an entropy model for the temperature field in the aluminothermic reduction magnesium smelting and studied the influence of heat transfer, thickness and furnace diameter on the entropy of the temperature field. It was concluded that the entropy of the temperature field is inversely proportional to the radius of the reduction tank and proportional to the height of the reduction tank and the heat transfer coefficient of material in the furnace. Using ferrosilicon alloy, Al-Si-Fe alloy and aluminum powder as reducing agents, Wenxin Hu et al. [87] prepared magnesium with vacuum thermal reduction and studied the burning loss rate, hydration activity and burning reduction of pellets with different particle sizes (1~12 mm) calcined at 600~1120 °C for 30~110 min. It was concluded that the calcium oxide slagging reaction in the process of the silico-thermal method reduces the critical temperature of reduction by 600 K and avoids the loss of MgO. Vacuum conditions can reduce the critical reaction temperature. The dissolution rate of alumina in the reducing slag CaO·2Al_2_O_3_ and CaO·Al_2_O_3_ reached 86%, and the total aluminum dissolution rate reached 88%. The sodium aluminate solution was carbonated to prepare aluminum hydroxide, and the whiteness reached 97%. The best calcination parameters of the different raw materials are shown in Table 4.

Chengbo Yang [88] prepared bulk magnesium and magnesium powder using self-designed, multi-stage condensation equipment (as shown in Figure 10). The results showed that the magnesium vapor is supersaturated after entering the condensing zone. As the temperature decreased, the magnesium collection efficiency increased, the binding rate of the growth interface increased, the dissociation rate decreased and the crystal growth rate increased. When the temperature gradient in the condenser is too large, the magnesium agglomerates and directly condenses below the liquefaction temperature. When the condenser temperature and temperature gradient are the same, the increase in magnesium partial pressure accelerates the crystal growth rate, and the increase in magnesium vapor concentration can reduce the oxidation rate of magnesium. Reducing the temperature gradient can weaken the phenomenon of uneven surface diffusion caused by rapid growth, improve the crystal quality and reduce the oxidation rate.

Wenxin Hu et al. [89] used Al-Si-Fe alloy as the reducing agent for the Pidgeon method to obtain higher magnesium vapor pressure during the reduction process, and the temperature required for the reaction was reduced. Using a thermodynamic calculation, they concluded that under the condition of a vacuum degree of 4 Pa, Al-Si-Fe alloy has a higher reduction efficiency of 66.8%, shorter reduction time, lower reduction temperature of 1373 K and longer service life. Yaowu Wang et al. [90] used the magnesite vacuum aluminothermic reduction method to produce magnesium and obtained a magnesium aluminate spinel by-product. Compared with the Pidgeon process, the ratio of raw materials to crude magnesium in the aluminothermic reduction process was 3.2~3.4:1, the consumption was reduced by 50%, the calcination energy consumption was reduced by more than 70%, the productivity was increased by 100%, the energy saving was more than 50%, the CO_2_ emission was reduced by up to 60% and zero waste residue emission is achieved. The process flow is as follows: firstly, magnesite is calcined at 850 °C, then the calcined magnesite powder and aluminum powder are mixed and pressed to make balls, and finally, the pellets are placed in a reduction furnace. The bedroom vacuum furnace used in the reduction experiment is shown in Figure 11. It was further concluded that the aluminothermic reduction reaction has the advantages of effectively prolonging the service life of the reduction tank and reducing the maintenance cost due to the low temperature required. Moreover, the aluminothermic reduction process has the characteristics of a high heat transfer coefficient and fast reaction rate, which shortens the reduction time and reduces energy consumption. The reduction slag can be used to produce a high-value-added refractory MA spinel [90].

Restricted by the principle, the Pidgeon process has the disadvantages of large consumption of raw materials, high energy consumption and large carbon emissions, and it is difficult to reduce the energy consumption and carbon emissions of the Pidgeon process [74]. Under the conditions including a reduction temperature of 1200 °C, reduction time of 2 h, excess coefficient of 5% aluminum powder and briquetting pressure of 30 MPa, NaiXiang Feng et al. [91] used the mixed minerals of dolomite and magnesite as raw materials (the mass ratio of material to magnesium was 3.1~3.3:1) to produce magnesium with the vacuum aluminothermic reduction, and prepared aluminum hydroxide by reducing slag. The results showed that the reduction rate of magnesium can reach more than 89%. The main phase of the slag was CaO·2Al_2_O_3_, and the content of Al_2_O_3_ was 67%. The leaching rate of alumina can reach more than 85% in the form of sodium aluminate in CaO·2Al_2_O_3_ alkaline solution. Compared with the Pidgeon method, the ore consumption was reduced by 50%, the energy consumption of the reduction process was reduced by more than 45% and the magnesium reduction rate was increased by more than 5%. Thus, comprehensive utilization of the reduction slag was realized. Qingfu Guo [92] used magnesite as raw material to carry out an aluminothermic reduction magnesium smelting test. It was concluded that compared with the silico-thermic method, the reduction temperature was reduced by 50 °C, and the energy consumption per ton of magnesium was reduced by about 80~90%.

### 2.3. Carbothermic Method

The magnesium smelting method is the premise and foundation for the development of the magnesium industry because the existing smelting method of magnesium metal generally has problems of high energy consumption, serious pollution, long processing and high cost. Therefore, it is very important to study the production of magnesium metal using the vacuum carbon thermal reduction of magnesium oxide. Compared with the existing magnesium smelting methods, this method has the characteristics of low energy consumption, low cost and low environmental pollution. Researchers at home and abroad have made good progress on the reduction process parameters, kinetic analysis, reactor modeling and low-carbon energy savings related to the carbon thermal method of magnesium refining.

Hamed A N et al. [93] investigated the effect of different carbon sources (carbon black, activated carbon and charcoal) on the performance of the carbon thermal reduction process. The ranking order of reduction conversion was obtained as activated carbon < carbon black < charcoal. The porous structure of charcoal provides better surface contact between dolomite powder and carbon and is environmentally friendly and less costly. Thermogravimetric experiments using dolomite and carbon black powder mixtures yielded that dolomite was calcined to MgO·CaO in the temperature range of 600~850 °C and then reacted with carbon in a reduction reaction at 1100 °C. In the presence of sufficient carbon, Mg was completely reduced before CaO began to be reduced. Adrian Coray et al. [94] investigated the reaction mechanism of magnesium refining using the carbothermal reduction process. At 1375~1450 °C and 1~2 KPa, MgO·CaO calcination produced CO, and then CO_2_ was generated as an intermediate product to reduce MgO and produce Mg_(g)_. It was further verified that Mg_(g)_ has two production pathways: (1) MgO dissociation reaction by heat and (2) a MgO reduction reaction with CO. Approximately twice as much Mg_(g)_ was produced from pathway (1) as pathway (2). J Puig et al. [95] prepared renewable Mg-Al nanostructured powders using the concentrated solar thermal reduction of MgO and Al_2_O_3_ at 1000~1600 Pa with argon as a carrier gas. They collected 100~250 mg of powder in the deposition region, with total metal yields of 40~52 wt% and powder metal contents of 60~80 wt% for the reaction. Yun J et al. [96] treated industrial magnesium oxide, metallurgical coke and calcium fluoride with ball milling and analyzed the effect of mechanical grinding on the rate of chemical reactions during metallurgy. They concluded that mechanical grinding has the advantages of increasing the contact area of reactants, enhancing the stability of adsorption between reactants and destroying the surface of magnesite grains to weaken graphite crystallization, which can significantly increase the reaction rate of magnesium oxide. The mechanism for the ball milling effect were also explained, including the three stages of magnesium oxide and coke in the ball milling process. In the first stage, ball milling reduces the particle and grain size, destroys the grain surface, forms surface defects and allows the material to mix uniformly. In the second stage, the particle size and crystallization remain significantly reduced, while the grain size remains stable. In the third stage, the grains do not change in size or surface (the size has reached the nanometer level), the contact area expands and a strong adhesion between the coke flakes and magnesite fragments is formed and aggregation occurs to form large particles. All three stages have an enhancing effect on the carbon thermal reduction process. Teng Zhang et al. [97] investigated the effect of mechanical activation on the vacuum carbothermal reduction of magnesium and concluded that CaF_2_ could break the [Mg=O] chemical bond compared with the carbothermal reduction of magnesium without mechanical activation. They also concluded that mechanical activation could reduce the grain size of CaF_2_, increase the specific surface area, increase the dislocation density, enhance the activity and mechanical energy storage, reduce the activation energy of the reaction and accelerate the reaction rate. When the reduction conditions were 1273 K and 1000 Pa, the effect of CaF_2_ on the vacuum carbon thermal reduction reaction showed a tendency to increase sharply and then stabilize with increasing mechanical activation time. Cheng-bo Yang et al. [98] investigated the behavior of Mg_(g)_ and CO_(g)_ during the vacuum carbon thermal reduction of magnesium oxide. The Mg vapor condensation process has an inverse reaction, producing carbon and MgO, which prevents the two metal clusters from combining with each other. When the vacuum pressure was 60 Pa, the supersaturated magnesium vapor caused the magnesium atoms to collide and agglomerate with each other, the Mg_(g)_ condensed directly into the solid phase at temperatures below liquefaction, and the concentration of CO was found to affect the quality of the condensed magnesium. Increasing the concentration of magnesium vapor reduced the inverse reaction rate of magnesium and resulted in a condensed specimen with a dispersed structure and low metallic luster. When the condensation zone temperature was 810~910 K, so that the temperature gradient at ΔT_3_ = 0.5 K/mm was the smallest, the inverse reaction rate between Mg_(g)_ and CO_(g)_ was lower, which is conducive to reducing the heat loss and increasing the liquid phase nucleation rate and magnesium vapor concentration. By controlling the condensation temperature, the inverse reaction rate of Mg can be reduced to improve the crystal quality, avoid rapid solidification and prolong the fusion time of Mg clusters. Yang Tian et al. [99] studied the role of CaF_2_ in the process of the vacuum carbothermic reduction of magnesium. In the MgO-C system (1500 K and 50 Pa), magnesium was generated with carbothermic reduction, and CaF_2_ was not involved in the carbothermic process. The mass loss increased with the increase in CaF_2_ content. After adding 5%CaF_2_, the reduction degree was obviously improved. The purity of the magnesium was 95.59 wt%, with a perfect crystal and lamellar structure. The mechanism for the reaction process in the vacuum carbothermic reduction of magnesium smelting was further studied [100], and it was found that no gas–solid reaction occurs between MgO and CO at 1723 K and 30~100 Pa. The main reduction reaction under the vacuum condition was MgO_(s)_ + C_(s)_ = Mg_(g)_ + CO_(g)_, and magnesium was obtained in the condensation zone. Due to the reverse reaction, a small amount of magnesium oxide appeared in the condensation product. Yun Jiang et al. [101] added 1%, 3% and 5% CaF_2_ to the raw materials in the vacuum carbothermal reduction method, and the reaction rate of the samples was about 26%. Calcium fluoride combined with magnesium oxide and silicon dioxide to form a eutectic, and the melting at 1573 K formed a channel that was conducive to the solid–solid reaction between carbon and magnesium oxide. Molten calcium fluoride provides free calcium ions and fluoride ions. Fluoride ions enter and distort the lattice of magnesium oxide, reducing its structural strength and chemical stability, which is conducive to the reduction of magnesium oxide with carbon. Calcium ions can be used to generate calcium silicate and magnesium silicate. Wei-dong Xie et al. [102] studied the morphology of magnesium samples condensed under different vacuum degrees and found that the magnesium at the bottom of the condensing cover was a feathery crystal with a close-packed needle structure and metallic luster. The top was a flaky crystal with a compact cluster structure. The metallographic structure is shown in Figure 12. The experimental results showed that the reduction rate of the carbothermal reduction reaction increases with a decrease in the vacuum degree, an increase in the reduction temperature and an extension in the reduction time. Furthermore, the reduction rate reached 83.7% under the conditions of 10 Pa, 1573 K and a reaction time 60 min. When the reaction temperature was in the range of 1423~1573 K, the reduction reaction rate constant k increased with the increase in temperature.

Yun Jiang et al. [103] used coke, charcoal and graphite for the carbothermal reduction of magnesium and analyzed the influence of the structure and composition of carbon materials on the reduction effect. The experiments showed that both charcoal and coke have higher reduction rates than graphite due to the presence of a large amount of amorphous carbon in the structure. It was further concluded that controlling the reduction temperature below 1700 K and ball milling before coke use can avoid the graphitization of amorphous carbon and increase the reduction reaction rate. Jie Chen et al. [104] mixed dolomite with carbon to make pellets and completed a new process of thermal decomposition and thermal reduction after one loading. When the vacuum degree was 10 Pa, the highest critical thermal decomposition temperature was 754.38 K, the critical reduction temperature of MgO was 1353.95 K the critical reduction temperature of CaO was 1531.41 K. These results were 369.38 K, 765.26 K and 897.26 K lower than that in atmospheric environment, respectively. Jiacheng Gao [105] studied the catalytic effect of calcium fluoride in the carbothermal reduction process. The reduction effects of four carbonaceous reductants were found as coke > charcoal > activated carbon > graphite. The optimum process conditions of the carbon thermal reduction of MgO with coke as a reducing agent were found to be a reduction time of 120 min, a reduction temperature of 1600 °C, a carbon ratio of 1:3 and a carbon particle size of 240 mesh. With the increase in CaF_2_ content, the reduction rate of MgO increased obviously, and the reduction rate increased greatly at lower temperature. The reduction rate of MgO increased from 67% to 99% at 1500 °C × 1 h and a CaF_2_ mass fraction of 3%. The reduction rate of MgO increased to 91% at 1400 °C × 1 h and a CaF_2_ mass fraction of 5%. Thus, adding CaF_2_ can accelerate the reaction and reduce the reduction temperature. Teng Zhang et al. [97] studied the effect of mechanical activation on the vacuum carbothermic reduction of the magnesium smelting process. The experimental results showed that the mechanical activation energy significantly enhances the vacuum carbothermal reduction process. With the increase in mechanical activation time, the weight loss rate in the sample increased. The mechanical activation energy greatly increased the dislocation free energy of magnesium oxide, thereby increasing the mechanical energy storage and reducing the activation energy of the reaction, so that the carbothermal reduction of magnesium metal could be carried out rapidly at 1273 K and 1000 Pa. With an extension in the mechanical activation time, the average grain size of magnesium oxide decreased, and the specific surface area of the reaction increased, which promoted the rapid reaction. Yaowu Wang et al. [106] studied the effect of CaF_2_ on the aluminothermic reduction of dolomite and magnesite. They found that CaF_2_ can effectively reduce the activation energy of the reduction reaction, accelerate the reduction reaction rate, and increase the reduction rate of MgO by more than 3%, but the effect of CaF_2_ decreased with the increase in the reaction temperature. Part of CaF_2_ participates in the reduction reaction to generate 11CaO·7Al_2_O_3_·CaF_2_. Another part of CaF_2_ is reduced by aluminum to form crystals of MgF_2_ and Al on the crystallizer. With an increase in the reduction temperature and the reduction reaction, most of the 11CaO·7Al_2_O_3_·CaF_2_ will be decomposed into CaO·2Al_2_O_3_ and CaF_2_. Thus, when the temperature exceeded 1100 °C, CaF_2_ was reduced by aluminum to form MgF_2_ and Al ash in the high temperature zone in the front of the mold. The promoting effect of CaF_2_ on the aluminothermic reduction reaction is the result of the combination of chemical reaction promoting and improving the surface activity of calcined white. Fa-ping Hu et al. [107] studied the Gibbs free energy and critical reaction conditions in the carbothermal reduction of magnesium and calcium. Based on the critical temperature and vacuum degree, a new thermodynamic criterion for the reduction reaction was proposed: when T/P 0.0449 < 1199.2, MgO and CaO cannot be reduced by carbon; when T/P 0.0449 ≥ 1199.2 and T/P 0.0462 < 1350.9, MgO can be reduced, but CaO cannot be reduced; and when T/P 0.0462 ≥ 1350.9, both MgO and CaO can decrease. Boris A. Chubukov et al. [108] studied the rate of the carbothermal reduction of MgO at 1350~1650 and 0.1~100 KPa from a kinetic point of view. When the conversion was less than 20%, the reduction rate increased with a decrease in pressure. When the conversion rate was 20~35%, the reduction rate decreased with a decrease in pressure, and the maximum reduction rate was at 10 KPa. At low conversion (α < 20%), the gas partial pressure inside the pellets was low, and the reduction rate was dominated by the solid–solid reaction between C and MgO. As the reaction proceeded, C and MgO were degraded by the reaction and sintering, and the reaction was dominated by a gas–solid reaction. Thus, under vacuum conditions, a relatively high mass transfer rate limits the MO_x_ + CO→MO_x−1_ + CO_2_ and C + CO_2_→2CO reactions. Tang Qifeng et al. [109] carried out a thermodynamic analysis on the carbothermal reduction process and studied the role of fluoride salts in the reduction process. The results showed that the starting temperature for the carbothermal reduction of MgO can be as high as 1900 °C under atmospheric pressure. Reducing the vacuum degree was found to effectively reduce the initial temperature of the reduction reaction, i.e., at 100 Pa, the starting temperature was 1230 °C and at 10 Pa, the starting temperature was 1100 °C. The reduction rate of MgO was 17.91% when no catalyst was used under the conditions including a vacuum degree of 2~5 Pa, reduction temperature of 1400 °C and reduction time of 15 min. The reduction rate of MgO was 67.05% at 1500 °C for 60 min. Fluoride salt was found to accelerate the reaction rate of carbothermal reduction. When the reduction time was 30 min, the catalytic effect of CaF_2_ was more significant. Kang Li et al. [110] carried out a thermodynamic and kinetic analysis on the behavior of magnesium oxide during vacuum carbothermal reduction. It was found that the MgO reduced by CO can be neglected under vacuum conditions due to the small amount of CO. The high temperature had little effect on the structure of MgO. The [Mg=O] bond did not break to form Mg vapor and oxygen, and there was no interaction noted for [Mg=O]. At 1723 K and 50 Pa, the MgO structure was not affected by high temperature, and the [Mg=O] bond was not broken. Yang Tian et al. [111] studied the reaction process of magnesium smelting with carbothermic reduction under vacuum conditions. The schematic diagram of the vacuum furnace equipment used is shown in Figure 13. It was concluded that the process is mainly divided into two stages: a reduction stage and a condensation stage. At 1723 K and 30~100 Pa, the main reduction reaction is MgO_(s)_ + C_(s)_ = Mg_(g)_ + CO_(g)_. In the condensation process, the condensation quality of magnesium is related to the concentration of CO, and C is formed after the condensation of magnesium vapor and inhibits the binding of metal droplets.

The behavior of Mg_(g)_ and CO_(g)_ in the vacuum carbothermal reduction method was experimentally investigated and simulated by Neng Xiong [112]. According to density functional theory, the interaction between a single CO molecule and Mg_(100)_ is physical adsorption. During the interaction between two CO molecules and Mg_(100)_, C_1_ and C_2_ form a strong [C=C] bond, and the [C_1_=O_1_] bond and [C_2_-O_2_] bond are stretched. Three CO atoms combine to form [C=C=C], and O atoms are released from the Mg_(100)_ surface. It was found that Mg_(g)_ condensed with CO_(g)_ to form MgO and carbonaceous aggregates using a SEM analysis. The formation of magnesium oxide in condensed magnesium vapor promoted the formation of a carbon chain. The formation mechanism of MgO was then divided into two processes using a dynamic simulation and experiment. Firstly, the following reaction occurs when multiple carbon monoxide atoms combine, nCO(g)→Mg[C=C=C]_n_ + nO, which is followed by Mg_(g)_ + O = MgO_(s)_. Illias Hischier et al. [113] independently designed a new experimental device to study the condensation/oxidation of metal vapor (as shown in Figure 14) and studied the vacuum condensation behavior of Mg(g) in CO(g) and CO_2_(g). When the temperature was 1000 °C, Mg_(g)_-CO_2(g)_ and Mg_(g)_-CO_(g)_ mixtures were used for the experiment, and the gas mixture flowed through the air-cooled tube condenser. When CO_2(g)_ was present in the environment, the Mg_(g)_ was rapidly oxidized; however, when there was CO_(g)_ in the environment and the temperature was higher than 950 °C, the Mg_(g)_ was not oxidized. This indicated that when the temperature is low, CO_(g)_ undergoes a disproportion reaction to oxidize Mg_(g)_, and with the decrease in CO_(g)_ partial pressure, the yield of metal Mg increases. Under the conditions of P_CO_ ≈ P_Mg_ = 0.6~13.2 mbar, T = 200~700 °C and a CO partial pressure of 90 w%, the condensation reaction rate of Mg_(g)_ was found to be much faster than the oxidation reaction rate.

Tao Qu et al. [114] redesigned the vacuum carbothermic reduction furnace, as shown in Figure 15, so that vacuum coking, vacuum thermal reduction and magnesium vapor condensation are completed in the same equipment, the distance between the reduction chamber and the condensation chamber is reduced. In this furnace, the temperature in the condensation chamber is increased, and the magnesium bar with high metal content is condensed. On this basis, a kilo-scale, medium-sized vacuum magnesium smelting furnace was designed and manufactured. As shown in Figure 16, the design of the condenser was changed from top-end to the side-end condensation, and magnesium was condensed in the middle of the extraction pipe. The temperature is controlled near the melting point of magnesium (649 °C), and a too-low temperature will cause magnesium vapor to condense into powder. Black ash (magnesium powder, oxide and Mg_2_C_3_ produced by condensation) is present in the agglomerates. The equipment greatly improves the reduction rate and direct yield of magnesium.

Hamed A N et al. [115] used dolomite as the raw material for the carbothermal reduction process. Comparing the solar carbothermal reduction magnesium smelting process with the traditional fire carbothermal reduction magnesium smelting process, they concluded that the most energy-consuming step is the carbothermal reduction of oxides. Thermogravimetric analysis showed that MgO in lime (MgO·CaO) was completely reduced at 1600 °C under normal pressure. This indicated that the carbothermal reduction of MgO·CaO can realize the gradual symbiosis of magnesium and calcium. The temperature and the ratio of MgO·CaO to carbon affected the production of magnesium and calcium. Furthermore, the electrothermal reduction experiments using different raw materials showed that the reduction rate of MgO·CaO is faster than that of MgO. Boris A. Chubukov et al. [116] compared the semi-system and the whole system carbothermal reduction method to produce metal magnesium, with the objective to reduce carbon emissions and energy consumption. Schematic diagrams for the two systems are shown in Figure 17. Under the conditions of 0.5 KPa pressure, a mixture of carbide, oxide and steel as the medium and using equimolar C/MgO particles as the feed to replace the gasified particles in the bed, magnesium and CO were mixed and evaporated with the semi-system vacuum carbothermal reduction method. The condensation rate of Mg_(g)_ was obtained (>85%), and the mass yield was M_Mg_ = 29.96 ± 0.13 g/h. In the temperature range of 1400~1550 °C, the mass yield of Mg(g) was M_Mg_ = 12.5~115.1 g/h. It was further concluded that the magnesium and reduction products that accumulated in the condenser promote the reduction reaction. The condensing plate must be made of a material with a high surface area to facilitate heat and mass transfer and shorten the residence time of gas and solids. The yield from the reduction reactor was limited by the heat transfer of the bed diameter. When the bed diameter was greater than 10 cm, the treatment capacity of the reactor was only limited by heat transfer. Wide, directly heated reduction furnaces can overcome these limitations.

### 2.4. Electrolytic Magnesium

Magnesium smelting with the thermal reduction method has problems including expensive reducing agents, expensive reduction tanks, a short service life, low single tank loading, low thermal efficiency, low mechanization degree and low production efficiency. Researchers have focused on the low-energy and environmentally friendly electrolytic magnesium smelting process, and very good progress has been made on the reduction process parameters, kinetic analysis, innovative process routes, reactor model establishment and industrial equipment. The preparation of magnesium and magnesium aluminum alloys with low temperature electrolysis can reduce energy consumption and help to protect the environment.

Fuxing Zhu et al. [117] used a mineral dissociation analyzer (MLA), scanning electron microscope (SEM), EDS and other means to analyze the existence form and source of impurity elements in magnesium metal produced with electrolysis. The results showed that the impurity phases in magnesium products have MgO, a Si (or Al) acid salt phase, an iron oxide phase, carbon-containing substances and electrolytes due to the oxidation of liquid magnesium, mass transfer of the lining materials in the electrolytic cell and refining furnace, dissolution of the iron container, oxidation of magnesium and the reaction of the electrolyte system. The purity of magnesium produced with electrolysis was found to reach 99.97% by controlling the temperature in the electrolytic cell and refining furnace at 660~690 °C in addition to controlling the inert gas protection and refining time of refining furnace (<60 min). Bin Liu et al. [118] analyzed the physical and chemical properties of brucite and magnesite and conducted carbon chlorination experiments. It was concluded that magnesium oxide calcined from brucite had the characteristics of small fragmentation tendency, high strength and strong wear resistance. When used as a furnace charge, it had the advantages of uniform air permeability and increased utilization of reaction space. The characteristics of the two raw materials during the thermal decomposition process in the chlorination furnace are shown in Table 5.

During the thermal decomposition of magnesite, the fine particles after decomposition increase the resistance of the material layer, and the particle size of the material is seriously dispersed, resulting in uneven permeability of the furnace material. The yield of magnesium chloride is reduced because the reactants cannot be fully contacted in the reaction space. During actual industrial production, some of the fine particles generated by decomposition are distributed in the decomposition zone, and the speed of mass transfer into the reaction zone is slow, so that the resistance of the material layer is maintained at a high level, resulting in an extremely uneven reaction in each part of the material layer. After the decomposition of brucite, the particles in the material layer are uniform, and the permeability is consistent. The yield of magnesium chloride is obviously improved because the reactants can make full contact in the reaction space, and the resistance of the material layer is small. G.M.RAO [119] studied the influence of fluoride content and electrolysis time on the current efficiency of magnesium electrolysis. The experiment showed that when fluorite is not added, magnesium leaves the cathode in a small ball shape and has poor cohesiveness. After adding fluorite to the electrolyte, the cohesion was obviously improved. The electrolysis efficiency was found to increase with an increase in fluorite concentration. When the fluorite concentration was 1~1.4%, the current efficiency increased up to 84%. However, the electrolysis efficiency decreased with the increase in fluoride concentration. By comparing the current density of 1 A/cm^2^ and the electrolysis time of 2 h, the current density of 1 A/cm^2^ and the electrolysis time of 8 h and the current density of 0.4 A/cm^2^ and the electrolysis time of 16 h, it was concluded that with short-term electrolysis, the current efficiency was the highest when the fluorite concentration was 1~1.4%. Zhe Sun et al. [120] carried out cold model experiments on the flow characteristics of chloride molten salt, liquid magnesium and high temperature chlorine in a magnesium electrolytic cell at high temperature according to the similarity criterion. The size ratio of the organic glass container used in the cold model to the industrial electrolytic cell was 1:5. According to the dynamic viscosity and density relationship, silicon oil and water were selected to simulate high temperature molten salt and liquid magnesium. The experiment showed that the order of the factors affecting the oil phase recovery rate φ in the cold model is: the inclination angle in the magnesium guide groove θ > the polar distance ACD > the electrode insertion depth D > the width of the magnesium collecting chamber W. The results showed that the collecting rate of magnesium increased with the increase in the gas flow rate, but it was basically affected by the bubble size. The recovery rate of the oil phase was 91.34% under the conditions of 60 mm electrode spacing, 28 mm electrode insertion depth, 127 mm width of magnesium collecting chamber and a 12° inclination angle in the magnesium channel. Jinzhong Chen et al. [121] determined the solubility of MgO in the electrolyte system MgCl_2_-NaCl-KCl-CaF_2_ (MgF_2_/NaF) and MgCl_2_-NaCl-KCl-NdCl_3_ using orthogonal tests and response surface analysis, and they studied the effect of each component in the molten salt on solubility. It was concluded that MgO is one of the main impurities affecting the electrolysis process of magnesium chloride. When the content of MgO in the electrolyte was more than 1%, the electrolysis current efficiency dropped sharply. In the MgCl_2_-NaCl-KCl-CaF_2_ (MgF_2_/NaF) system, the content of MgCl_2_ and CaF_2_ was found to be the main factor affecting the solubility of MgO, but the solubility of MgO in the system was CaF_2_ > NaF_2_ > MgF_2_. In the MgCl_2_-NaCl-KCl-NdCl_3_ system, MgO had a large solubility. The content of NdCl_3_ directly determined the solubility of MgO, and the solubility at 780 °C satisfied MgO_(%)_ = 21.46W_3_ − 0.1964. Thermodynamic studies have shown that the interaction between magnesium oxide and the components in the molten salt and the formation of complex ions are the main reasons for the dissolution of MgO in the molten salt. Zhiguang Zhang et al. [122] studied a new process for the production of metal magnesium with molten salt electrolysis using low-sodium carnallite from the Qarhan Salt Lake area of Qinghai Province as the raw material. Firstly, the low sodium carnallite was dehydrated once to remove the attached water and part of the crystal water. Then, the primary dehydrated potassium carnallite powder was subjected to secondary melting to remove the remaining 5% crystal water. Then, anhydrous potassium carnallite was added to the electrolytic cell, which was electrolyzed into Cl_2_ and crude magnesium, and then refined in a continuous refining furnace. Yaron Aviezer et al. [123] studied a new process for producing magnesium metal from seawater, and the process flow diagram is shown in Figure 18. Mg^2+^ was extracted from seawater with an ion exchange membrane and impurities such as Na^+^ and K^+^ were filtered. The extracted magnesium reacted with ferrosilicon and CaO to form a Mg(OH)_2_·Ca(OH)_2_·FeSi solid filter cake. The filtrate of the raw cake was mainly composed of CaCl_2_, which was used as ion exchange regeneration solution. Ca^2+^ was then adsorbed on the resin and supplemented with CaO to form Mg(OH)_2_ precipitates. Mg(OH)_2_ contacted with FeSi in the dry cake, and Mg^2+^ was dehydrated and thermally reduced in a 1150 °C retort. Overall, Mg with a purity of 99.0~99.5% was obtained with the feasibility dehydration reduction test. The cost of the proposed process was CNY 5190 per ton of magnesium higher than that of the Pidgeon process.

Hangjin Wu et al. [46] prepared magnesium at 10 Pa pressure using calcined dolomite and ferrosilicon as raw materials in an electrothermal coupling field. The experimental results showed that the electrothermal coupling field magnesium smelting process can effectively reduce the reduction temperature required for magnesium smelting with the traditional pyrometallurgical process. As the reaction time increased, the DC electric field strength decreased, the current density increased and the reduction rate increased significantly. Under the conditions of a DC electric field intensity of 950 V/cm, equipment temperature of 700 °C and current density of 1.18 A/cm^2^, the reduction rate of 150 min was 88.35%. The purity of magnesium obtained with X-ray diffraction and electron probe microscopy was 98.54 wt%. Under the blackbody radiation model, the Joule heating effect increased with the increase in the current density. As the reaction proceeded, the reduction rate increased due to the increase in sample resistance due to the consumption of Si and MgO. Xiaoyun Hu et al. [124] used molten salt direct electrochemical deoxidation (FFC) to prepare metal magnesium with low temperature molten salt electro-deoxidation. The principle of cathode reduction test system is shown in Figure 19.

The mechanism of low temperature molten salt electrolysis of solid magnesium oxide was determined with a graphite cavity electrode. It was concluded that in the cathodic reduction process, partial magnesium oxide was first electrolyzed to generate magnesium and O^2−^, then O^2−^ and Ca^2+^ reacted with unreduced MgO to generate CaMg_3_O_4_ and CaMg_6_O_7_, which was then further reduced to generate magnesium, CaMg_6_O_7_ and CaMg_3_O_6_. The generated CaMg_6_O_7_ continues to be reduced to metal magnesium and Ca_3_Mg_3_O_6_. Ca_3_Mg_3_O_6_ was eventually reduced to magnesium. When the sintering temperature was too high, the molten salt was not easily diffused into the interior and the deoxidation reaction process was slow; however, when the sintering temperature was too low, MgO was easy to pulverize. When the sintering time was too long, the molten salt had difficultly penetrating into the sintered sheet, which slowed down the reaction rate of the electrolysis process. Finally, the sintering temperature of MgO was determined to be between 1100 °C and 1200 °C, the sintering time was 3~6 h, and the molding pressure was 20 MPa. With the increase in molten salt temperature and CaCl_2_ content, the electrolysis reaction rate increased. The working electrode, reactor and cell of the FFC method are shown in Figure 20.

M Korenko et al. [125] proposed an annual output of 34,000~36,000 t/year from the MgO electrolytic magnesium smelting process. The schematic diagram and experimental device are shown in Figure 21. The electrochemical kinetics and mass transfer parameters were simulated with thermodynamic and finite element numerical models. Solar heat input reduced the energy costs from USD 0.654/kg to USD 0.481/kg, but the magnesium production from electrolyzers also decreased, thus increasing the equipment costs and maintenance costs. The cost of producing tons of magnesium is about CNY 17,023. Compared with the current commercial process for producing magnesium, it has better economic advantages, and the CO_2_ emission level is 46% lower than that of the Pidgeon process.

Longjiao Wang et al. [126] independently designed a small, multi-stage electrolytic cell. Using the new electrolyte ratio and the extreme vertex design method to select the measurement points, the physical properties of the salt system in the multi-pole electrolytic cell and the influence of the change in each component in the electrolyte system on the physical properties of the electrolyte were studied, as shown in Table 6. The regression equation for the initial crystal temperature, density, surface tension and conductivity of the salt system in the multipole electrolytic cell was obtained using their experiment, as shown in Equations (13)–(16).
(13)$$y_{T}=2616.21x_{1}+936.52x_{2}+522.18x_{3}-9109.26x_{1}x_{2}-1051.85x_{1}x_{3}+1109.26x_{2}x_{3}$$
(14)$$y_{\rho }=2.82x_{1}+2.79x_{2}+1.65x_{3}-2.91x_{1}x_{2}-1.46x_{1}x_{3}-0.91x_{2}x_{3}$$
(15)$$y_{F}=3052.32x_{1}+433.68x_{2}+301.31x_{3}-4986.67x_{1}x_{2}-4606.66x_{1}x_{3}-225.00x_{2}x_{3}$$
(16)$$y_{\sigma }=8.86x_{1}+9.82x_{2}+3.89x_{3}-27.59x_{1}x_{2}-7.41x_{1}x_{3}-14.63x_{2}x_{3}$$

During the electrolysis process, the electrolysis zone I is filled with the generated chlorine gas, which carries the magnesium droplets upward. The mixture of electrolyte, magnesium and chlorine flows through the electrolysis chamber, multi-electrode and box cathode in turn. When the diameter of the cathode was reduced, the chlorine gas and the mixed liquid were fully separated and entered the IV zone above the electrolysis chamber due to the turbulence formed by the flow direction turning up to 90°. Under the action of the exhaust fan, it entered the exhaust gas treatment device and was consumed by the sodium hydroxide solution. After the electrolyte and magnesium mixture flow into the box cathode, part of the metal magnesium floats above the liquid and enters the magnesium collecting chamber in the II area of the figure along the magnesium conducting tank. The remaining magnesium is carried into the magnesium collecting chamber by the electrolyte through the magnesium conducting hole on the partition wall. The electrolyte enters the bottom of the electrolytic cell with the large circulation of the electrolytic cell, flows through the electrolyte buffer zone III composed of the support plate and the support, and enters the electrolytic cell through the rectangular flow hole on the support plate for electrolysis. The structure of the multi-stage electrolytic cell and the circulation path of the electrolyte are shown in Figure 22. The addition of fluorite into the chloride molten salt electrolyte can reduce the freezing point and viscosity of the melt and act as a flux to remove the oxide film on the metal surface to increase the yield of magnesium.

### 2.5. Relative Vacuum Method

In the context of energy conservation, emission reduction and environmental protection, the Pidgeon process for magnesium smelting can only be used intermittently due to the switching of vacuum and atmospheric pressure. After calcination in the rotary kiln, 5% of the fine powder is produced, resulting in a great waste of energy and resources. CO_2_ emissions are large, i.e., for the production of 1 t metal magnesium, there are emissions of about 23 t CO_2_ (excluding ferrosilicon smelting). Professor Zhang Ting-an led a team [127,128] who broke through the theoretical limitations of the vacuum concept and put forward the concept of the ‘Relative vacuum’, which realized the magnesium smelting process under normal pressure or slightly positive pressure. The so-called relative vacuum is the difference between the actual partial pressure of magnesium vapor in the container and the equilibrium partial pressure of the reaction at this temperature. The actual partial pressure of the gas is lower than the equilibrium partial pressure by carrying the inert gas to achieve a certain relative vacuum. From the point of view of reaction thermodynamics, the equilibrium partial pressure of magnesium vapor is determined at a certain temperature, and the reaction can be carried out only when P_Mg,R_ < P_Mg,E_. The higher the reaction temperature, the larger P_Mg,E_ and the larger ∆P, where the definition ∆P is relative vacuum. Among them, ∆P is the relative vacuum, P_Mg,E_ is the equilibrium partial pressure of magnesium, P_Mg,R_ is the actual partial pressure of magnesium in the reactor, P_V_ is the total pressure in the reactor, P_atm_ is atmospheric pressure and K is the reaction equilibrium constant of magnesium. The process uses prefabricated pellets as raw materials. The reduction process is carried out continuously under relative vacuum conditions, and the obtained metal magnesium is continuously condensed into liquid. Because there is no need to switch between vacuum and atmospheric pressure, the continuous production of magnesium metal is realized, the vacuum system and vacuum reduction tank are cancelled, the defects of single heat transfer mode and low heat transfer efficiency of Pidgeon method are avoided, the convective heat transfer is increased, the reduction cycle is greatly shortened, the whole process automation can be realized and the flowing carrier gas can be reused. The self-developed continuous magnesium smelting process is shown in Figure 23. The prefabricated pellets (raw materials for vacuum magnesium smelting) are calcined followed by in situ reduction with no need to cool after the ball and other operations. The device shown in Figure 24 was developed to realize the calcination and reduction of pellets in one equipment. Compared with the Pidgeon method, the rotary kiln calcination equipment is cancelled, which can better utilize the waste heat of flue gas and calcined white and significantly improve the energy utilization efficiency. The mineralization of CO_2_ utilization significantly reduces carbon emissions in line with the dual-carbon policy.

Magnesium is a high vapor pressure metal. When the temperature reaches the melting point, the vapor pressure is 350 Pa. After exceeding the melting point, the vapor pressure of magnesium rises sharply, reaching 101.3 KPa at 1380 K. During the reduction reaction, the partial pressure of magnesium vapor in the condenser must be greater than the saturated vapor pressure of the condenser at the same temperature in order to condense into magnesium metal. The condensation collection process of magnesium vapor in the silicon-thermal method is simple, and there is no reverse reaction. Condensed magnesium can be obtained by reasonably controlling the temperature and pressure in the condensing collection area. According to the definition of a relative vacuum, the actual partial pressure of magnesium is lower than its equilibrium partial pressure, and the reduction reaction can occur. According to the principle of thermodynamics, the relationship between temperature and equilibrium partial pressure of magnesium vapor can be obtained, as shown in Equation (17) and in the image shown in Figure 25. The equilibrium partial pressure increases with increasing temperature. In order for the reaction to occur, the partial pressure of the product gas must be lower than its equilibrium partial pressure. The flowing inert gas is used to carry the gas of the product away from the reaction system, so that the partial pressure of the product gas is lower than the equilibrium partial pressure of the reaction, that is, in the ‘Relative vacuum’ state, the reduction reaction is promoted.
(17)$$\left\{\begin{array}{c} lgP^{*}=-7780/T-0.855lgT+13.535\left(298K\sim 923K\right)\\ lgP^{*}=-7750/T-1.411lgT+14.915\left(T>923K\right) \end{array}\right.$$

Assume that the reduction process conforms to the unreacted core model, as shown in Figure 26. The generation and migration of magnesium vapor need internal diffusion and external diffusion. Magnesium vapor is produced by the reduction reaction and then migrates to the surface of the pellet in the form of diffusion inside the pellet. The magnesium vapor on the surface of the pellet diffuses into the argon system through the gas film in the form of external diffusion, and the is finally carried away from the reaction zone by the flowing inert gas. Assuming that the reduction process is a quasi-interface chemical reaction process, the quasi-interface chemical reaction rate can be expressed as:(18)$$Rr=\frac{4\pi r_{d}^{2}k(P_{e}-P_{d})}{RT}$$
where *R_r_* is the quasi-interface chemical reaction rate, unit mol/s; *r_d_* is the radius of the unreacted core, unit m; *k* is the quasi-interface chemical reaction constant, the equilibrium partial pressure of the product gas at *P_e_* reaction temperature, unit Pa; and *P_d_* is the actual partial pressure of the product gas at the interface, unit Pa. The diffusion rate of the product vapor in the product layer can be expressed as follows:(19)$$R_{d}=\frac{-4\pi r_{}^{2}D_{e}}{RT}\frac{dP}{dr}$$
where *R_d_* is the diffusion rate of the product gas, unit mol/s; *D_e_* is the effective diffusion coefficient of the product gas in the product layer, unit m^2^/s; and *P* is the partial pressure of the product gas in the product layer with radius r, unit Pa. The effective diffusion coefficient of material in porous media can be estimated using the Bosenquet formula:(20)$$\frac{1}{D_{e}}=\frac{\tau }{\varepsilon }(\frac{1}{D_{k}}+\frac{1}{D_{g}})$$
where *D_k_* is the Kundsen diffusion coefficient, *D_g_* is the molecular diffusion coefficient of the product gas, *τ* is the pore curvature of the solid and *ε* is the porosity of the solid. The diffusion rate *R_g_* of the product gas in the film is expressed as:(21)$$R_{g}=\frac{-4\pi {\left(r_{0}+\delta \right)}_{}^{2}D_{g}}{RT}(\frac{P}{P_{Bm}})\frac{dP}{dr}$$
where *r*_0_ is the radius of the pellet, *δ* is the thickness of the gas film, *P* is the total pressure of the gas and *P_Bm_* is the logarithmic mean value of the pressure of the inert gas on both sides of the gas film, which can be calculated using Equation (22):(22)$$P_{Bm}=\frac{P_{B2}-P_{B1}}{\ln(\frac{P_{B2}}{P_{B1}})}$$
where *P_B_*_1_ and *P_B_*_2_ are the actual partial pressures of the inert gases on both sides of the gas film, respectively. Assuming that the reduction system is in a steady state, the gas production rate should be equal to its diffusion rate:(23)$$R_{r}=R_{d}=R_{g}=R$$

In the formula, *R* is the number of substances participating in the reaction per unit time, and the unit is mol/s. The definition is as follows:(24)$$R=\frac{dn_{g}}{dt}=-\frac{dn_{M}}{dt}=-\frac{d}{dt}(G\frac{4}{3}\pi r_{d}^{3})=-4\pi r_{d}^{2}G\frac{dr_{d}}{dt}$$

In the formula, *G* is the content of reduced oxide in unit pellet volume, unit mol/m^3^; *n_g_* is the molar number of the generated gas, unit mol; and *n_M_* is the total amount of reduced oxide in the unreacted core, unit mol. The kinetic relationship between the reaction rate and time can be obtained by combining the Equations (20)–(24). In the condenser, the high temperature gas flows along the cold wall, and a condensate film is formed between the high temperature gas and the cold wall. A temperature gradient and a magnesium vapor pressure gradient appear near the liquid film. For ease of discussion, this layer is called the gas film. The high temperature gas transfers to the liquid film through the gas film, and the magnesium vapor condenses into liquid magnesium, and the heat carried by it is transferred to the cooling medium through the gas film and the liquid film. Assuming that the cold wall temperature is the same at *T*_0_, the inlet temperature of the gas is *T*_1_ and the partial pressure of magnesium is *P*_1_. The outlet temperature is *T*_2_, and the partial pressure of magnesium is *P*_2_. In order to maximize the condensation of magnesium vapor into liquid, the following conditions must be met: (1) the cold wall temperature *T*_0_ is slightly larger than the melting point temperature of magnesium and (2) the gas outlet temperature is *T*_2_ = *T*_0_ and the magnesium partial pressure is *P*_2_ = *P*_T0_*, where *P*_T0_* is the vapor pressure of magnesium at temperature T_0_. Inside the gas film, heat and mass transfer occur simultaneously, while only heat transfer occurs inside the liquid film. The heat and mass transfer rate determine the temperature distribution in the gas film and liquid film, the pressure distribution of magnesium vapor in the gas film and the condensation rate of magnesium vapor. In general, the heat transfer rate is not equal to the mass transfer rate, and the condensation rate control step is slower. From the kinetic point of view, when the magnesium vapor particles produced with the reaction leave the reaction interface rapidly, the actual vapor pressure at the reaction interface is less than the equilibrium partial pressure. Obviously, the use of inert gas without a vacuum can make the actual partial pressure lower than the equilibrium partial pressure. When the flow rate of inert gas reaches a certain value, the actual partial pressure of magnesium at the chemical reaction interface can also be smaller than the equilibrium partial pressure at the chemical reaction interface. Compared with the traditional magnesium smelting technology, the advantages of this technology are significant: (1) the rapid continuous magnesium smelting method eliminates the vacuum system and vacuum reduction tank, which is a simple in operation with low in equipment requirements and reduces equipment investment and operating costs. (2) About 5% of the fine powder produced in the calcination process of magnesium smelting with the traditional silicon-thermal method cannot be used and wasted. The relative vacuum technology directly uses uncalcined dolomite to make pellets and then calcine pellets. There is no problem related to the waste of fine powder, and the resource utilization rate is significantly improved. (3) The low-temperature, fast calcination of dolomite is realized. The calcined pellets are continuously transported to a high-temperature reduction furnace for high-temperature reduction without cooling. The waste heat from the calcined exhaust gas and the waste heat from the high-temperature reduction exhaust gas are directly used to preheat the pellets and inert carrier gas. (4) In the calcination process of pellets, ferrosilicon reacts with calcium oxide to form silicon–calcium alloy, which is in the liquid phase at the reduction temperature, so that the reduction process is carried out between the solid and liquid phases, and the rapid and deep reduction of magnesium oxide is realized. This breaks through the technical bottleneck of slow mass transfer and low reduction speed caused by the solid–solid reaction in traditional silico-thermic magnesium smelting. (5) The continuous production of magnesium under non-vacuum conditions is achieved in a flowing inert argon atmosphere, which greatly shortens the reduction cycle. The magnesium reduction cycle is shortened from 10~14 h to 60~90 min. The recovery rate of metal magnesium is greatly improved, and the comprehensive recovery rate of metal magnesium is increased to more than 90%. The inert carrier gas can be recycled. (6) The CO_2_ produced in the calculation process of prefabricated pellets is captured and absorbed by calcium-containing alkaline waste to obtain ultra-fine calcium carbonate powder, which realizes the clean utilization of high-value min of CO_2_ tail gas.

## 3. Comparison of Parameters in the Different Magnesium Smelting Methods

Yanwen Yan et al. [129] proposed the ‘Ammonia leaching calcium-carbonation calcium fixation’ process to treat a large amount of calcium-containing solid waste produced with silico-thermic reduction of magnesium smelting. Using a NH_4_Cl solution as the leaching agent, the effects of leaching temperature, leaching time, the liquid–solid ratio, NH_4_Cl mass fraction and magnesium slag particle size on the leaching effect of calcium in slag were analyzed. The data are shown in Table 7.

It was concluded that when the temperature was in the range of 328~363 K, the mass fraction of NH_4_Cl was 30%, and the calcium leaching rate was 80%:p_(NH3)_/p_θ_ = 0.2, the Gibbs free energy ΔG of the Ca_2_SiO_4_ phase leaching reaction in magnesium slag was less than 0 and Ca_2_SiO_4_ could be decomposed into Ca^2+^ solution. The results showed that with an increase in the reaction temperature, a finer particle size of magnesium slag, an increase in the liquid–solid ratio and an increase in the NH_4_Cl mass fraction, the leaching effect of calcium will be enhanced. The leaching rate of calcium in magnesium slag was found to reach 77.01% under the conditions including a NH_4_Cl mass fraction of 30%, leaching temperature of 90 °C, liquid–solid ratio of 10 and leaching time of 4 h.

Kan Hong et al. [130] used calcium silicate slag as a raw material to recover tungsten and molybdenum with the alkali roasting-ball milling leaching process. The experiment showed that the process has strong applicability, and the recovery rate of tungsten and molybdenum can reach the ideal level when dealing with the solid solution slag of calcium silicate smelting with tungsten and molybdenum. Using Na_2_CO_3_-Na_2_O_2_ roasting at 800 °C, a Na_2_CO_3_ dosage of 50% of the raw material weight, a Na_2_O_2_ dosage of 10% of the raw material weight and roasting 60 min after adding water ball milling, they measured a tungsten molybdenum leaching rate of more than 92%. In addition, the further influence of the tungsten molybdenum leaching rate was found in the order: roasting temperature > Na_2_CO_3_ usage > roasting time. Guangxiang Ji et al. [131] mixed magnesium slag as a mineral admixture into cement concrete to study the effect of magnesium slag content and fineness on the working performance, mechanical properties and durability of concrete. The experimental results showed that magnesium slag can reduce the early strength of concrete due to its retarding effect on hydration, but it could also enhance the later strength. At the same time, it was found that magnesium slag can optimize the long-term performance of concrete, reduce drying shrinkage and improve sulfate resistance and frost resistance. The fluidity of concrete increased with the increase in magnesium slag content, and the fineness of magnesium slag had little effect on fluidity. The optimum dosage was 30%, and the specific surface area was 485 m^2^/kg. The accumulation of cerium in natural water resources can lead to serious environmental problems. A.G. Morozova et al. [132] proposed the use of metallurgical slag calcium magnesium aluminum silicate (CaO-MgO-SiO_2_-Al_2_O_3_) to prepare a composite adsorbent to treat cerium (Ce^3+^) ions in waste liquid. The migration characteristics of calcium (Ca^2+^), magnesium (Mg^2+^), silicon (Si^4+^) and aluminum (Al^3+^) ions were studied using adsorption experiments with a high concentration Ce^3+^ aqueous solution (1000 mg/L). The removal rate of Ce^3+^ ions reached 93.36% within 24 h. Baoguo Fan et al. [133] used NaCl, CaCl_2_, Na_2_SO_4_, CaSO_4_, Na_2_CO_3_, K_2_CO_3_, acetic acid and adipic acid to modify the quenched hydrated magnesium slag to improve the desulfurization performance. The optimum desulfurizer was 2% Na_2_SO_4_ modified magnesium slag, and the calcium conversion rate reached 40.16%. It was found that Cl^−^ and SO_4_^2−^ could improve the desulfurization performance of magnesium slag by increasing the specific surface area and pore volume of the magnesium slag, and the modification effect of SO_4_^2−^ was better than that of Cl^−^. CO_3_^2−^ reacts with Ca(OH)_2_ to form CaCO_3_ on the surface of magnesium slag, which improved the desulfurization performance of magnesium slag. Organic acids promoted the transport of H^+^ in the hydration paste and promoted the dissolution of Ca_2_SiO_4_. Li Jia et al. [134] modified magnesium slag by adding BaO, B_2_O_3_ and P_2_O_5_ during the hydration and quenching process of magnesium slag. BaO, B_2_O_3_ and P_2_O_5_ inhibited the nucleation of γ-C_2_S, prevented the crystal phase transformation of β-C_2_S during cooling, and increased the content of highly active β-C_2_S. The highly active β-C_2_S is formed by hydration [C=S=H], which enhanced the physical adsorption capacity of magnesium slag. The increase in crystal structure defects enhanced the chemical adsorption capacity of magnesium slag. The improvement in the overall adsorption capacity of magnesium slag improved its desulfurization performance. Overall, the effect on desulfurization performance of chemical adsorption capacity was greater than the physical adsorption capacity. Li Jia et al. [135] carried out quenching and hydration experiments on magnesium slag under different conditions (quenching and hydration temperature, hydration time, liquid/solid ratio and continuous/discontinuous process). When the quenching temperature was 950 °C and the hydration temperature was 80 °C, the highest degree of hydration was 0.16. The optimum conditions for continuous hydration were determined: a quenching temperature of 950 °C, a liquid/solid ratio of 8 and a hydration time of 8 h. The order of influence for the three factors on hydration degree was: hydration time > quenching temperature > liquid/solid ratio. The desulfurization performance of the samples treated with continuous hydration was better than that of the samples treated with discontinuous hydration, and the calcium conversion rates were 30.3% and 13.3%, respectively. Liguang Xiao et al. [136] used magnesium slag to prepare cementitious materials and analyzed the influence of magnesium slag content and material grinding process on the strength of magnesium slag cementitious materials. It was concluded that when the content of magnesium slag and slag are equal, the magnesium slag cementitious material has better strength. The greater the content of cement clinker, the higher the strength of the magnesium slag cementitious material. Compared with the ‘grinding before mixing’ process, the magnesium slag cementitious material prepared with the ‘mixing before grinding’ process had better strength. Yuming Tian et al. [137] added magnesium slag to the ceramsite proppant and concluded that the magnesium slag does not affect the phase of the ceramsite proppant. At high temperatures, the magnesium slag formed a liquid phase with the oxide in the raw material, making the ceramsite have better mechanical properties. After adding magnesium slag, the sintering temperature of ceramsite was reduced, which saves fuel and alleviates the environmental pollution caused by magnesium slag. The commercial production of magnesium has been ongoing for more than one hundred years, and the diversity in the production process is one of the important characteristics of the magnesium industry. The current magnesium metal production technology is mainly divided into two categories: magnesium chloride molten salt electrolysis and thermal reduction. A comparison of the two is shown in Table 8, which is based on the representative silicon-thermal Pidgeon method. The reducing agent is 75% ferrosilicon and the raw material is dolomite. During the electric heating method without direct emissions of CO_2_, fossil energy power generation accounted for 65% of the total converted into direct emissions of CO_2_ power generation.

It can be seen from the table that after considering the raw materials, energy, reaction conditions, carbon emission, energy consumption, material magnesium ratio, production cost, magnesium yield and magnesium purity of magnesium smelting, the aluminothermic reduction method is superior to the carbothermic reduction method, and the silico-thermic reduction method is superior to the aluminothermic reduction method [138,139,140,141,142,143]. Table 9 shows the data from a comparison of the reaction conditions, reduction rate, energy consumption and carbon emission of different reducing agents. It is concluded that the silicon- thermal method is superior to the aluminothermic method and carbothermal method because of the easy availability of reducing agents and good economy. It is worth noting that the ‘Relative vacuum’ continuous magnesium smelting process breaks through the thermodynamic limitations of the original technology that must be produced in a vacuum environment. The ‘Relative vacuum’ theory based on the equilibrium vapor pressure of the metal realizes the rapid continuous smelting of metal magnesium under the actual micro-positive pressure ‘Relative vacuum’ rather than ‘Absolute vacuum’ conditions. The reduction cycle is shortened from 10~14 h in the Pidgeon method to 1~1.5 h, and the comprehensive recovery rate of magnesium reaches more than 90%. There is also a reduction in the per ton of magnesium metal consumption of standard coal from 5 t to 3~3.5 t and CO_2_ emissions from 22 t to 11~13 t. The solution for the current magnesium smelting method cannot be continuous production or labor-intensive with heavy pollution and low production efficiency [144,145,146,147,148]. The industrial induction vertical jar device for magnesium smelting greatly improves the service life of the reduction jar, solves the problems of slow heat transfer and long heating time and lays an industrial foundation for the continuous production of magnesium metal.

Magnesium smelting with molten salt electrolysis has become the main method of producing metal magnesium because of its low cost and wide source of raw materials. The metal magnesium produced by it accounts for about 75% of the total magnesium production. Magnesium chloride, sodium chloride, potassium chloride and other mixed molten salts are used for electrolysis. Magnesium chloride is continuously consumed as a raw material for extracting magnesium, while other chlorides can improve the physical and chemical properties of the electrolyte. Before electrolysis, magnesium chloride must be dehydrated, and the decomposition potential of water is lower than that of magnesium chloride, resulting in waste of energy and damage to the electrolytic tank. Anhydrous magnesium chloride, according to different raw materials, is divided into magnesite (or magnesium oxide) chlorination production of anhydrous magnesium chloride and a magnesium chloride solution to produce low water magnesium chloride hydrate and anhydrous magnesium chloride. The process includes solution purification, concentration, drying and dehydration. At present, the most representative processes are the DSM process, Dow process, Magnola process, magnesium hydroxide process, AMC process and MagCorp process. The parameters and processes of various electrolysis processes are compared in Table 10.

## 4. Application and Development Prospect of Magnesium and Magnesium Alloys

Under the policy of energy conservation, emission reduction and environmental protection, the comprehensive energy consumption per ton of magnesium in magnesium smelting enterprises is controlled below 4.5 t standard coal. The consumption of dolomite is less than 10.5 t, the consumption of ferrosilicon (Si > 75%) is less than 1.05 t and the consumption of fresh water is less than 10 t. The recovery rate of magnesium is higher than 80%, the utilization rate of silicon is higher than 75%, and the recovery rate of crude magnesium refining is higher than 96%. The comprehensive utilization rate of reduction slag is higher than 70%. The high energy consumption of molten salt electrolysis limits its application range. In the thermal reduction method, the aluminothermic reducing agent is expensive, and the carbon thermal method seriously pollutes the environment. Although the silico-thermal method is widely used in industry, it is limited by vacuum conditions and cannot be continuously produced.

Compared with the Pidgeon process, the consumption of standard coal per ton of magnesium in the vacuum continuous magnesium smelting process is 3~3.5 t, which is reduced by more than 33.5%. The reduction cycle is shortened from 10–14 h to 1–1.5 h, and the carbon emission per ton of magnesium is reduced from 23 t to 11–13 t. Most importantly, breaking the limitation of vacuum conditions, continuous production can be achieved, and magnesium production can be increased by more than 55%. The CO_2_ in the calcination process is absorbed by calcium-containing alkaline waste slag, and the reduction slag can be used as refractory material to achieve zero solid waste, low energy consumption and low carbon emission. This provides a new way for green metallurgy.

## Figures and Tables

**Figure 1 materials-16-03340-f001:**
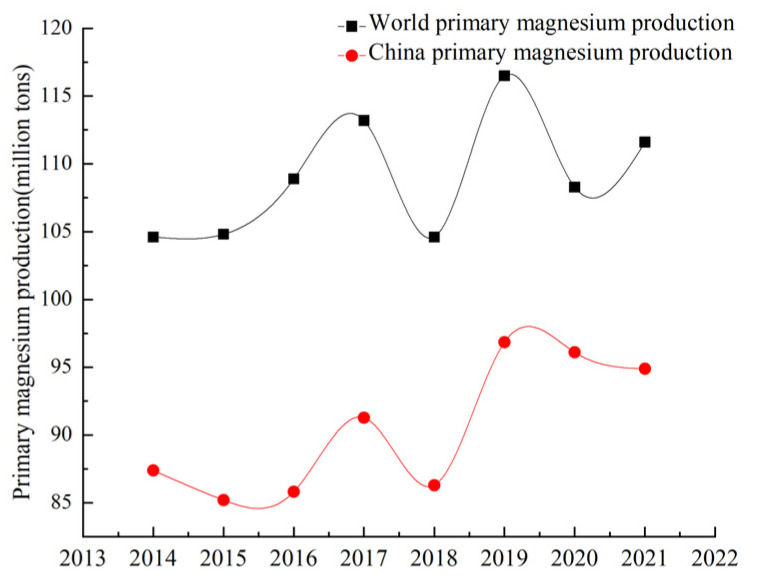
China and the world’s raw magnesium production trend.

**Figure 2 materials-16-03340-f002:**
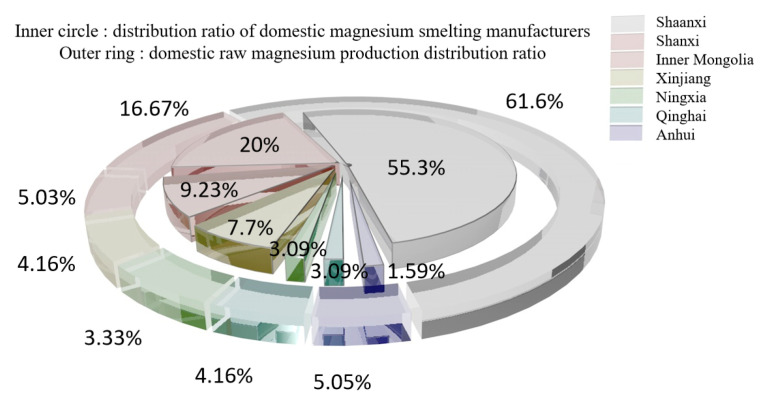
Distribution of magnesium smelter and raw magnesium production in 2020.

**Figure 3 materials-16-03340-f003:**
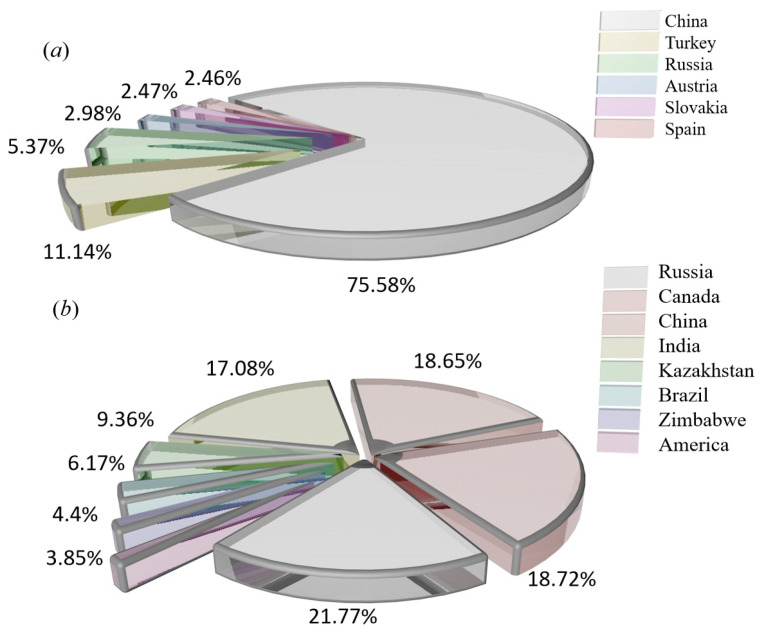
(**a**) Production histogram showing the world’s major magnesite producing countries. (**b**) Production percentage pie chart showing the world’s serpentine producing countries.

**Figure 4 materials-16-03340-f004:**
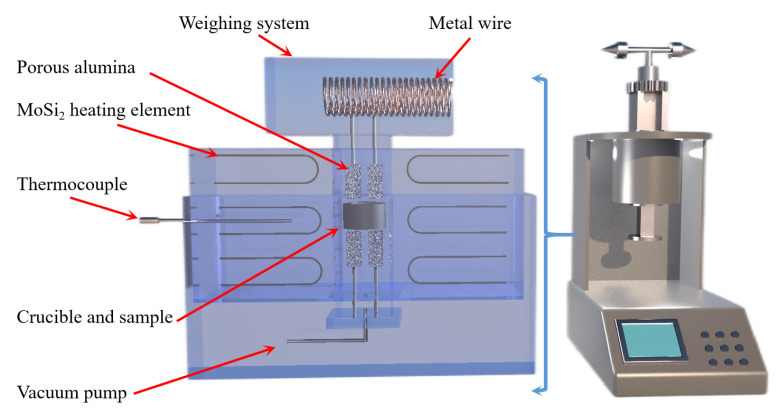
A thermogravimetric analysis (Zhengzhou University) schematic diagram and equipment model diagram.

**Figure 5 materials-16-03340-f005:**
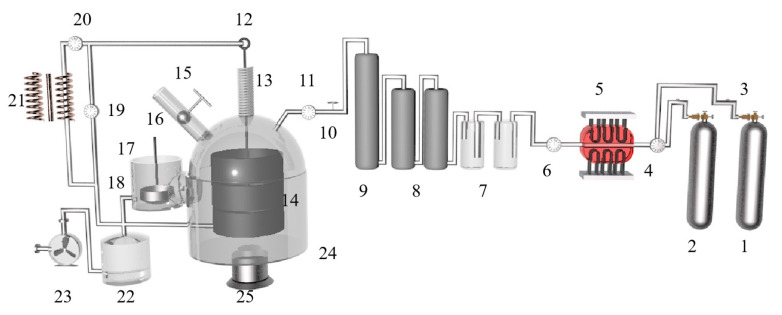
A principal diagram showing the magneto-thermal reactor: 1. argon cylinder, 2. hydrogen cylinder, 3. gas regulator, 4. double taps, 5. electric furnace, 6. double taps, 7. sulfuric acid bubbler, 8. silica tower, 9. silica tower, 10. inlet taps to the unit, 11. vacuum meter, 12. mobile graphite electrode of 3 cm diameter, 13. extensible plastic pipe, 14. fixed graphite crucible, 15. feeder, 16. heater, 17. condenser, 18. outlet tap, 19. vacuum meter, 20. amp meter, 21. welding transformer, 22. trap, 23. rotating vacuum pump, 24. lower movable opening and closing reaction chamber and 25. hydraulic jack.

**Figure 6 materials-16-03340-f006:**
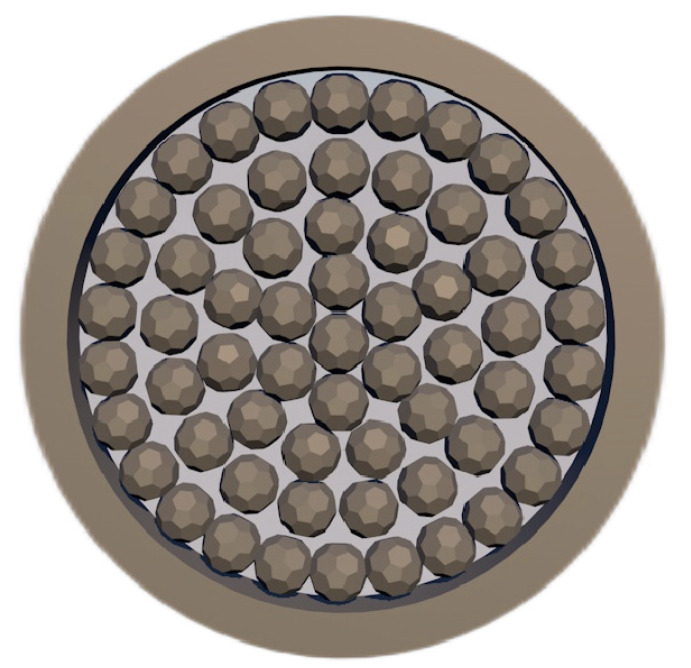
Arrangement of pellets in the reduction tank.

**Figure 7 materials-16-03340-f007:**
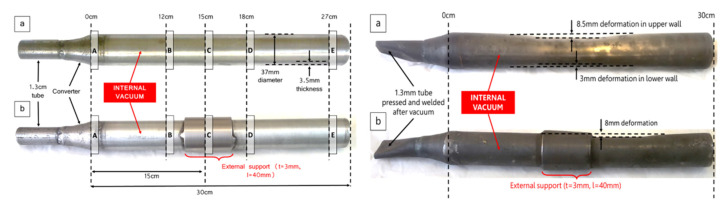
(**a**)—Unsupported and (**b**)—supported vulnerable tubes, creeping miniature distillers, exposed to 1200 °C and 10 Pa internal temperatures for 60 Days. The distance from B to A is 12 cm, from C to A is 15 cm, from D to A is 18 cm, from E to A is 27 cm.

**Figure 8 materials-16-03340-f008:**
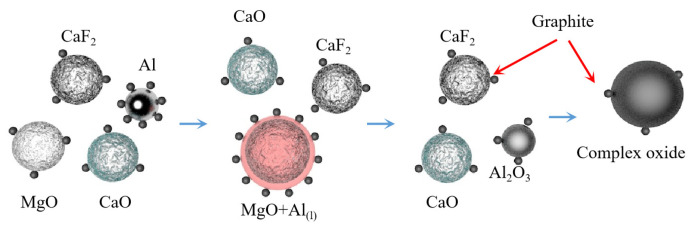
Reaction mechanism diagram.

**Figure 9 materials-16-03340-f009:**
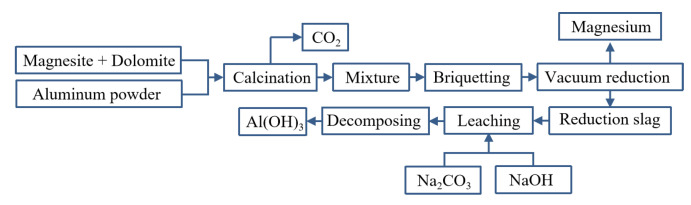
Aluminothermic process for magnesium smelting and the reduction slag treatment process.

**Figure 10 materials-16-03340-f010:**
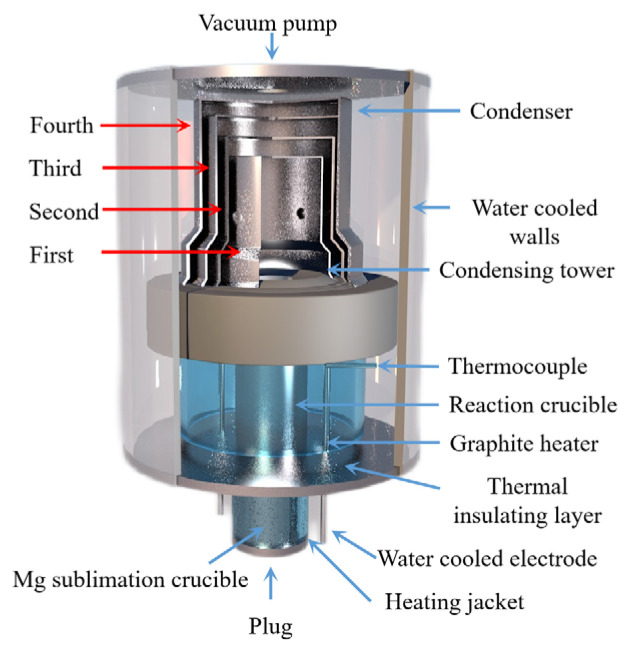
The Kunming University of Science and Technology self-designed, multi-stage condensing plant.

**Figure 11 materials-16-03340-f011:**
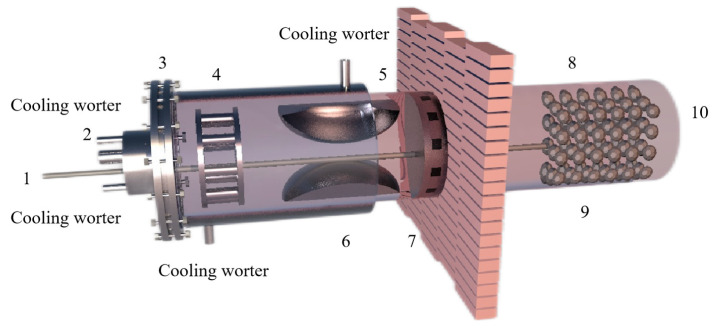
Principle diagram showing the vacuum thermal reduction tank: 1—thermocouple, 2—vacuum tube, 3—water cooling jacket, 4—sodium potassium condenser, 5—magnesium condenser, 6—rough magnesium, 7—heat cut-off, 8—coal hopper, 9—coal hopper and 10—furnace.

**Figure 12 materials-16-03340-f012:**
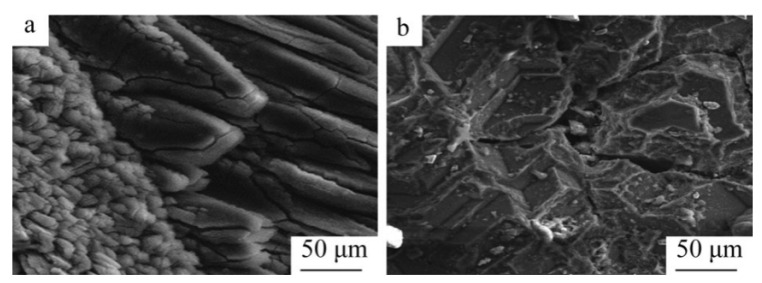
SEM images showing carbothermic reduction product samples crystallized in different sections of the condensation cap: (**a**) sample from the bottom section and (**b**) sample from the top section [59].

**Figure 13 materials-16-03340-f013:**
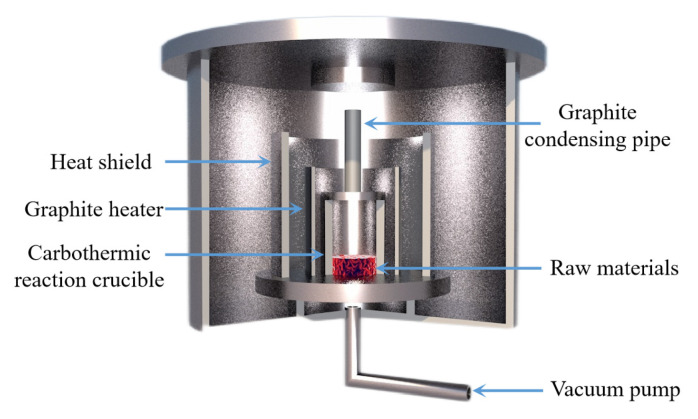
Principle equipment in a magnesium vacuum furnace for carbothermal reduction.

**Figure 14 materials-16-03340-f014:**
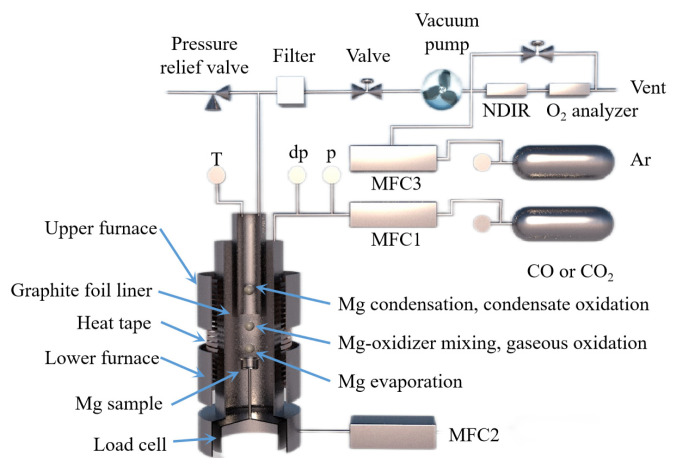
A schematic diagram showing a magnesium evaporation–condensation system with different regions: (1) magnesium evaporation, (2) magnesium-oxidizer mixing and gaseous oxidation and (3) magnesium condensation and condensate oxidation.

**Figure 15 materials-16-03340-f015:**
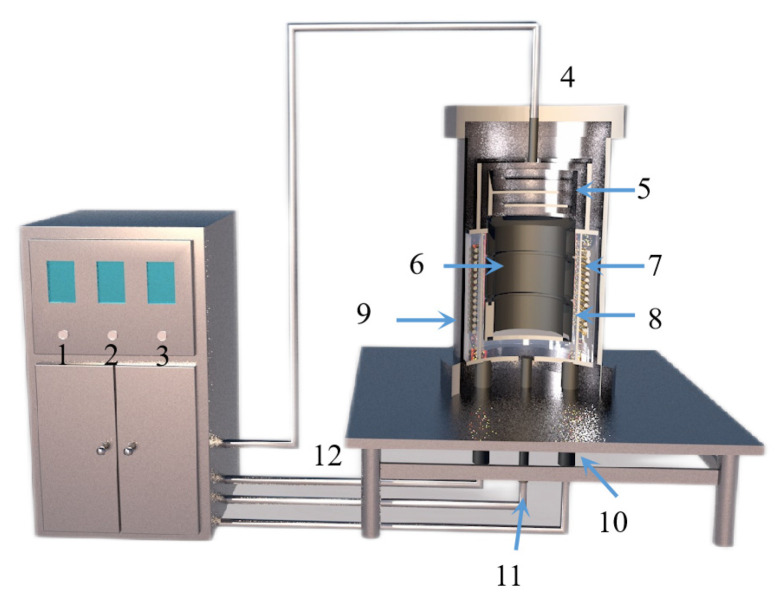
A structure diagram showing a small vacuum reduction experimental device composed of: 1. silicon-controlled temperature control power supply; 2. temperature display instrument; 3. mac vacuum gauge; 4. thermocouple; 5. condenser; 6. crucible; 7. heating elements; 8. insulation cover; 9. vacuum furnace body; 10. electrode; 11. thermocouples; and 12. exhaust pipe.

**Figure 16 materials-16-03340-f016:**
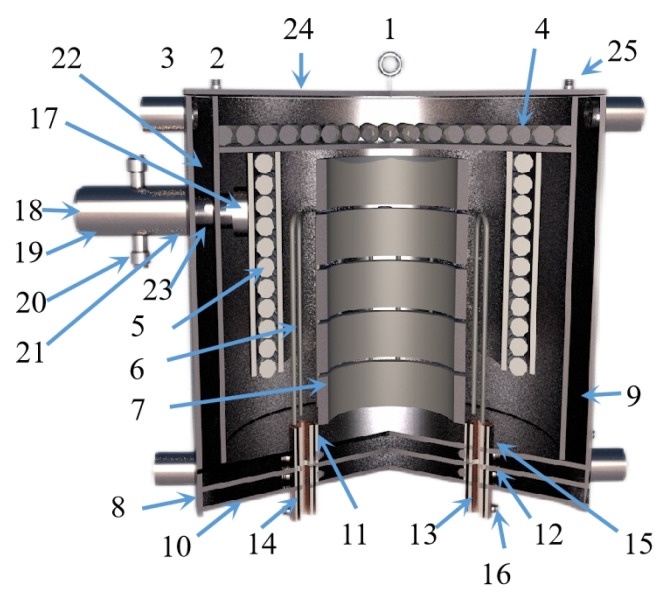
A diagram showing the Kg vacuum furnace structure composed of: 1. hanging ring; 2. cooling water inlet and outlet pipe; 3. cooling water jacket; 4. carbon felt insulation furnace lining; 5. graphite insulation reflective screen; 6. graphite heater; 7. graphite crucible; 8. stainless steel shell; 9. insulation refractory brick; 10. stainless steel base; 11. graphite conductive base; 12. rubber sealing gaskets; 13. water-cooled copper electrode; 14. rubber insulation sleeve; 15. graphite gasket; 16. electrode nut; 17. carbon material filter sieves; 18. condenser; 19. carbon felt insulation graphite cover; 20. vacuum pumping connection pipeline; 21. graphite casing; 22. graphite cover; 23. rubber sealing ring; 24. stainless steel furnace cover and 25. thermocouple combination.

**Figure 17 materials-16-03340-f017:**
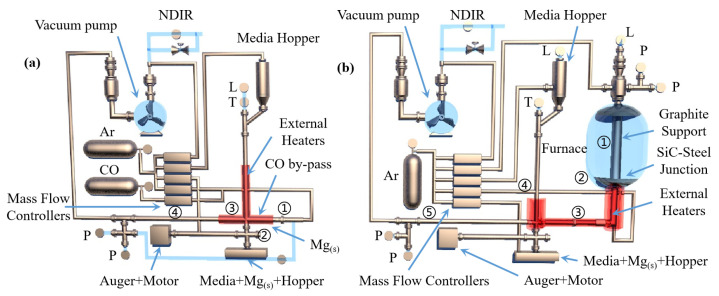
The measured values are ‘L’ = liquid level, ‘P’ = pressure, ‘T’ = temperature and ‘dP’ = differential pressure. (**a**) Semi-system: Mg_(s)_ is evaporated (1) to form Mg_(g)_ in a vacuum at 725 °C. At 850 °C, CO is mixed with Mg_(g)_ downstream of the evaporation zone (2) and flows into the moving bed (3) where Mg_(g)_ is deposited. CO is pumped through the moving bed, and the remaining Mg_(g)_ deposits on the outlet wall (4). (**b**) The whole system: equimolar C/MgO pellets are gasified in a vacuum furnace at a gasification temperature of ≥ 1400 °C (1). The gas product passes through a water-cooled seal (2), and then passes through a transmission line (3) and hits the steel medium in the moving bed condenser (4). CO is pumped through the moving bed, and the remaining Mg_(g)_ deposits on the outlet wall (5).

**Figure 18 materials-16-03340-f018:**
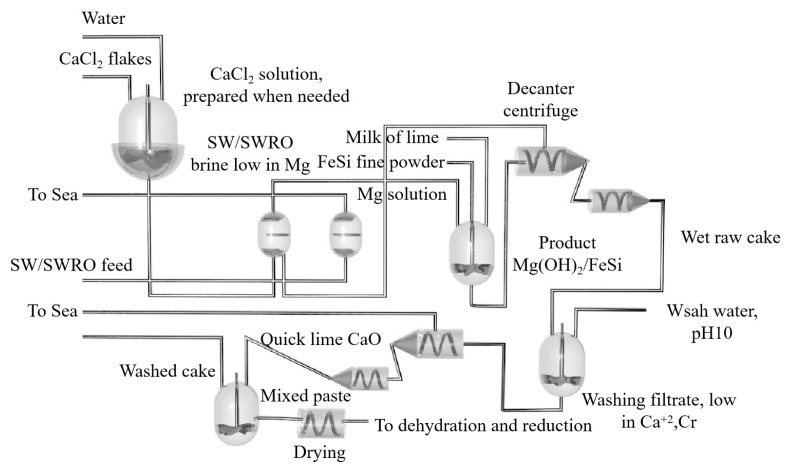
Process diagram showing materials in a thermal reduction tank including a Mg^2+^ separation step based on ion exchange, a Mg(OH)_2_ precipitation step, centrifugal filtration, filter cake washing, lime addition and drying.

**Figure 19 materials-16-03340-f019:**
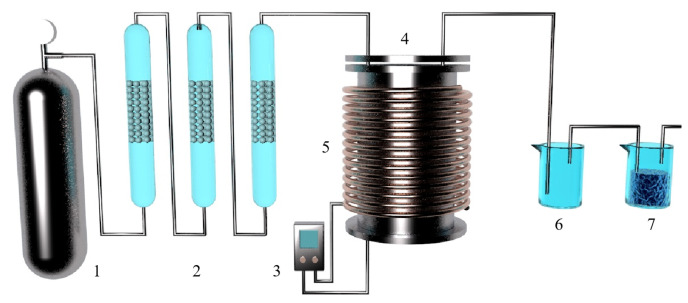
A diagram showing the electrolysis experimental system device including 1. Ar gas cylinder; 2. gas drying system; 3. WZK temperature controller; 4. reactor; 5. tap resistance furnace; 6. buffer bottle and 7. tail gas absorption bottle.

**Figure 20 materials-16-03340-f020:**
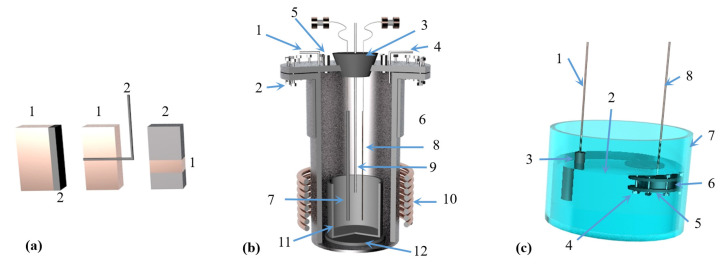
(**a**) Working electrode composed of a 1 metal body and 2 oxide layer used in the FFC method. (**b**) A schematic diagram showing a reactor composed of 1 inlet 2 sealing ring 3 sealing plug 4 outlet 5 window 6 cooling water 7 graphite anode 8 cathode wire 9 thermocouple 10 heating body 11 crucible 12 insulator; (**c**) A schematic diagram showing an electrolytic cell composed of 1 anode lead 2 mixed molten salt 3 graphite anode 4 screw 5 graphite sheet 6 magnesium oxide 7 corundum crucible and 8 cathode lead.

**Figure 21 materials-16-03340-f021:**
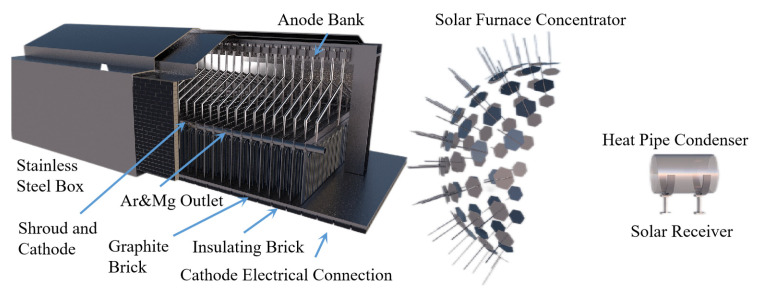
A 5 kW Solar Oven Heating Coupled Laboratory-Scale Magnesium Oxide Electrolytic Cell Solar Receiver with 0.75-Inch Inconel 625 Sodium Heat Pipe.

**Figure 22 materials-16-03340-f022:**
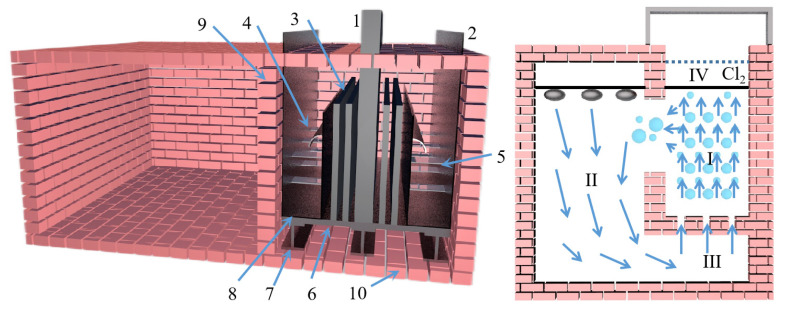
Electrolyte circulation diagram showing the 1. anode; 2. box cathode; 3. multi-electrode; 4. inverted magnesium tank; 5. guide plate; 6. support plate; 7. support plate; 8. electrolytic chamber cover; 9. electrolytic cell side wall and 10. tank bottom. I. electrolysis area; II. magnesium collection area; III. electrolyte buffer area and IV separation area.

**Figure 23 materials-16-03340-f023:**
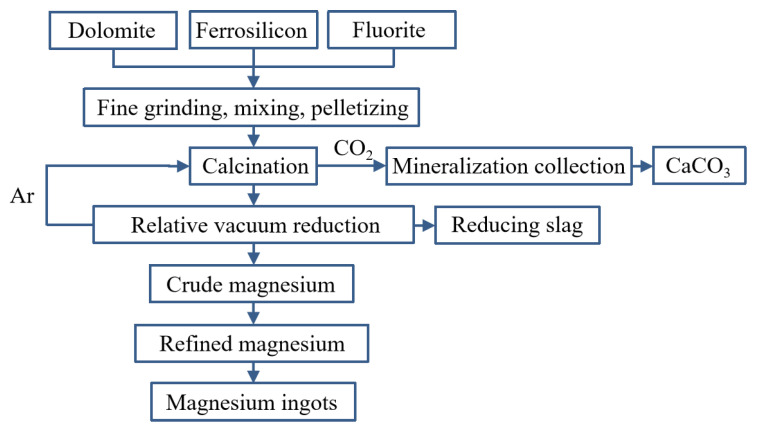
The relative vacuum continuous magnesium smelting process.

**Figure 24 materials-16-03340-f024:**
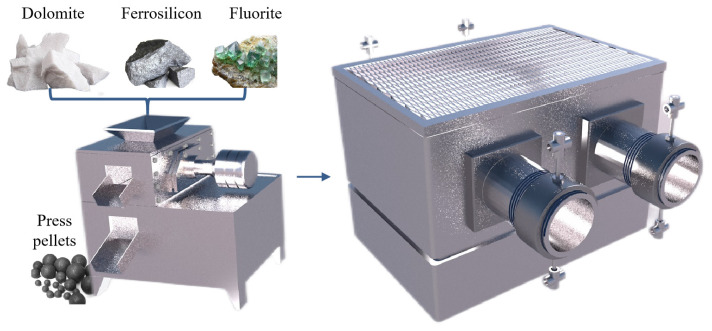
A relative vacuum calcination reduction furnace.

**Figure 25 materials-16-03340-f025:**
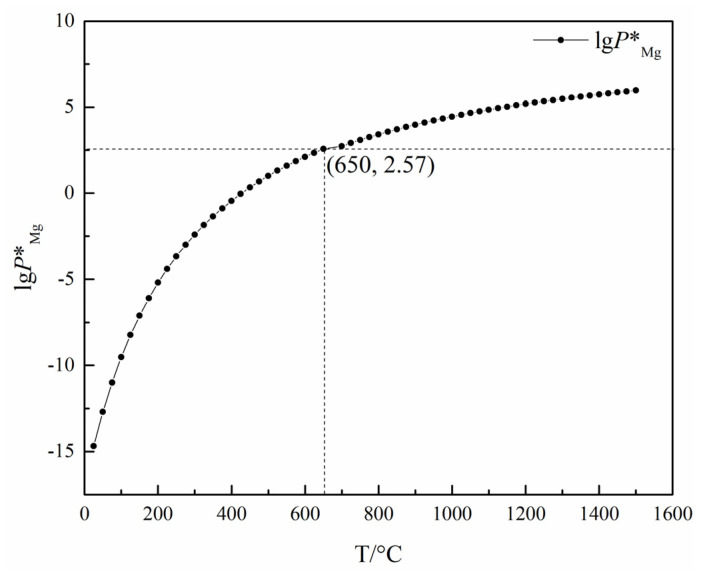
Equilibrium partial pressure of magnesium vapor at different temperatures.

**Figure 26 materials-16-03340-f026:**
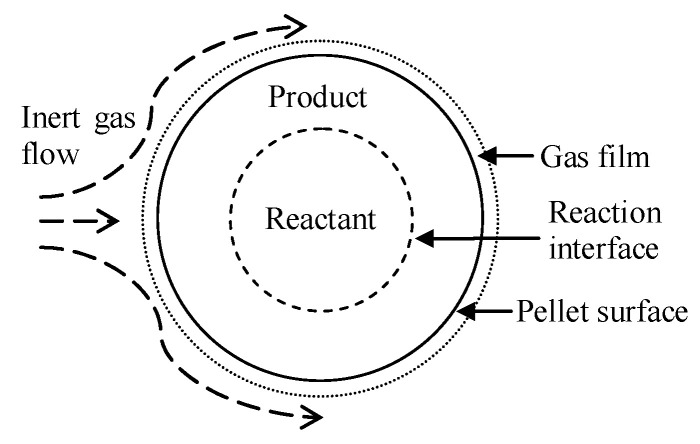
Kinetic model showing continuous reduction under vacuum condition.

**Table 2 materials-16-03340-t002:** Concentrations of main ions in seawater at different salinities [39,40,41].

Salinity	Na	Mg	Ca	K	Sr	B	Cl	SO_4_
15	4.662	0.56	0.179	0.173	0.003	0.002	8.37	1.173
20	6.239	0.749	0.238	0.231	0.005	0.003	11.202	1.571
25	7.827	0.94	0.299	0.29	0.006	0.003	14.055	1.97
26	8.147	0.978	0.311	0.302	0.006	0.003	14.630	2.051
27	8.467	1.017	0.323	0.314	0.006	0.003	15.205	2.132
28	8.788	1.055	0.336	0.325	0.007	0.004	15.779	2.212
28.72	9.018	1.083	0.344	0.334	0.007	0.004	16.193	2.271
29	9.108	1.094	0.348	0.337	0.007	0.004	16.354	2.293
30	9.428	1.132	0.360	0.349	0.007	0.004	16.929	2.374
31	9.749	1.171	0.373	0.361	0.007	0.004	17.506	2.454
31.36	9.865	1.185	0.377	0.365	0.007	0.004	17.714	2.484
32	10.072	1.21	0.385	0.373	0.007	0.004	18.083	2.536
33	10.393	1.248	0.397	0.385	0.007	0.004	18.663	2.616
33.5	10.5545	1.2675	0.4035	0.391	0.0075	0.004	18.9525	2.657
34	10.716	1.287	0.41	0.397	0.008	0.004	19.242	2.698
36	11.36	1.364	0.435	0.421	0.009	0.004	20.401	2.86
38	12.004	1.441	0.46	0.445	0.01	0.004	21.56	3.022
40	12.648	1.518	0.485	0.469	0.011	0.004	22.719	3.184
42	13.292	1.595	0.51	0.493	0.012	0.004	23.878	3.346

**Table 3 materials-16-03340-t003:** Evaluation of the solid reaction mechanism function of the kinetic equation.

Symbol	Form of Functions		Correlation R^2^	E (kJ·mol^−1^)	Log A (s^−1^)	n
	f(α)	g(α)				
F_1_	(1 − α)	−ln(1 − α)	0.99137	261.272	7.498	1
F_2_	(1 − α)^2^	(1 − α)^−1^ − 1	0.99247	287.222	8.622	2
F_n_	(1 − α)^n^	1 − (1 − α)^1−n^(1 − n)^−1^	0.99418	320.270	9.896	1.8
A_n_	n(1 − α)[−ln(1 − α)]^n−1/n^	[−ln(1 − α)]^1/n^	0.99131	250.112	7.063	1.035

**Table 4 materials-16-03340-t004:** Optimum forging parameters of dolomite, magnesite and magnesian dolomite [87].

Ore Types	Pellet Size	Calcination Temperature	Calcination Time	Calcined White Burning Loss Rate	Corresponding Hydration Activity
Dolomite	4 mm	1060 °C	90 min	46.56%	35.02%
Magnesite	12 mm	840 °C	90 min	51.03%	42.3%
Huntite	12 mm	880 °C	90 min	49.86%	40.68%

**Table 5 materials-16-03340-t005:** Thermal Decomposition Characteristics of Brucite and Magnesite [118].

Raw Material	Particle Shape after Thermal Decomposition	Particle Size Uniformity of Material Layer	Material Permeability	Contact Area between Reactants	Reaction Rate
Magnesium hydroxide	Essentially constant	Uniform	Good, uniform	Larger	Higher
Magnesite	Small particles, large quantity	Non-uniform	Non-uniform	Smaller	Lower

**Table 6 materials-16-03340-t006:** Physical Properties of the Salt System in a Multipole Electrolyzer and the Effect of Composition Changes in the Physical Properties of the Electrolyte [126].

Electrolyte Performance	Range	Effect of NaCl on System	Effect of MgCl_2_ on System	Effect of CaCl_2_ on System	Electrolysis Requirements
Liquidus temperature	599.29~666.12 (°C)	Increase as the content increases	Decrease as the content increases	Decrease as the content increases	Content not exceeding 56%
Density	1.694~1.733 (g/cm^3^)	Decrease as the content increases	Decrease as the content increases	Increase as the content increases	1.694~1.733 (g/cm^3^)
Surface tension	107.787~114.965 (μN/mg)	Increase as the content increases	Decrease as the content increases	Increase as the content increases	Increasing the proportion of MgCl_2_
Electrical conductivity	2.145~2.291 (S/cm)	Increase as the content increases	Decrease as the content increases	Decrease as the content increases	Increasing the proportion of NaCl

**Table 7 materials-16-03340-t007:** Table showing the factors that influence the leaching rate [129].

Influencing Factors	Leaching Temperature (°C)	Leaching Time (h)	Liquid–Solid Ratio	NH_4_Cl Mass Fraction (%)	Magnesium Slag Particle Size (μm)
At different levels (leaching rate)	60 (33%)	1 (37%)	6 (34%)	26 (43%)	150~270 (29%)
	70 (36%)	2 (55%)	8 (46%)	28 (48%)	96~150 (43%)
	80 (44%)	3 (71%)	10 (56%)	30 (54%)	74~96 (52%)
	90 (56%)	4 (77%)	12 (57%)	32 (56%)	<74 (56%)

**Table 8 materials-16-03340-t008:** Comparison of various magnesium smelting methods.

Comparative Content	Silico-thermic	Aluminothermic	Carbothermic	Relative Vacuum	Electrolytic Method
Raw material	Dolomite, Magnesite	Dolomite, Magnesite	Dolomite, Magnesite	Dolomite, Magnesite	Magnesite, Dolomite, Hydromagnesite, Carnallite, Serpentine, Seawater
Quality of raw material (MgO%)	23.8%_(Dolomite)_ 46.0%_(Magnesite)_	23.8%_(Dolomite)_ 46.0%_(Magnesite)_	23.8%_(Dolomite)_ 46.0%_(Magnesite)_	23.8%_(Dolomite)_ 46.0%_(Magnesite)_	——
Energy	Coal, Natural gas	Coal, Natural gas	Coal, Natural gas	Natural gas, Water gas, Electricity	Hydropower, Gas, Fuel
Reaction condition	10~20 Pa 1473~1503 K [47]	10~20 Pa 1443~1473 K [75]	30~100 Pa 1573~1667 K [96]	1.013 × 105 Pa 1573~1623 K [127,128]	1.18 A/cm^2^ >973 K [117]
CO_2_ produced with mineral calcination	5.284 t [50,52,55]	——	(Dolomite) 5.002 t (Magnesite) 2.210 t [106,116]	3.540 t [127,128]	——
CO_2_ produced with energy supply	22.135 t [58,59]	——	(Dolomite) 9.838 t (Magnesite) 6.060 t [97,106]	11~13 t [127,128]	——
Energy consumption per ton of magnesium production/tce	4.549 [128]	8.735 [79,80,83]	1.500 [94,98]	3.047 [127,128]	4.224~5.445 [46]
Single reactor yield/(kg·tank^−1^)	20~35 [71]	30 [86]	18~19 [112,113]	15~20 [127,128]	Single groove 0.9 t (24 h) [46]
Magnesium ratio	6.3~6.5 [70.72]	5.8~6.0 [91]	6.5~8.0 [106,115]	10~10.1 [127,128]	——
Energy Cleanliness	Fossil energy	Fossil energy	Fossil energy	Fossil energy, Clean energy electricity	Clean energy electricity, 35% green energy
Cost (Production cost per t/CNY)	20,970 [48]	21,567 _(3CaO·Al2O3)_ 19,015 _(CaO·2Al2O3)_ 19,631 _(CaO·Al2O3)_ 26,380 _(12CaO·7Al2O3)_ [75,76,79,79,81]	11,980 [100,102]	20,843 _(3CaO·Al2O3)_ 18,318 _(CaO·2Al2O3)_ 18,780 _(CaO·Al2O3)_ 25,445 _(12CaO·7Al2O3)_ [127,128]	16,520 [118,121]
Purity of crude magnesium	97.86~99.98% [52]	>99.90% [74]	>95.59% [103]	99.95% [127,128]	99.95% [123]
Magnesium extraction rate	65~87% [55]	85~94% [88]	>80% [97]	>80% [127,128]	78% [121,123]
Mechanical automation	Lower	Lower	Lower	Middle	Higher
Slag utilization rate	High MgO and Fe content, maximum addition of cement raw materials 9~12%	Recovery and reuse of calcium aluminate reduction slag as raw material for production of aluminum hydroxide with alkali leaching	Production of silicon calcium iron alloy, vacuum distillation preparation of calcium and silicon iron alloy	No solid waste generated	Chlorine-containing waste gas produced by the anode, to be discharged after chlorine adsorption by lime milk
Merit	Minimal investment, simple process, high purity magnesium, low cost	Low raw material consumption, high production efficiency, low energy consumption and no waste residue emission, the carbon emission in the production process can be reduced by 30~50%	Reduced energy consumption, reduced CO_2_ emissions, reduced waste generation	Pellet calcination and reduction in a device, better use of flue gas and calcined waste heat, significantly improve energy efficiency, greatly reduce CO_2_ emissions	Low cost, wide source of raw materials
Existing problems	Low thermal efficiency, high energy consumption, high pollution, high labor intensity, low unit reactor output, high cost of reduction tank	Large consumption of raw materials, high energy consumption, large carbon emissions	Thermal efficiency is not high, the work is intermittent operation, low production capacity	Not industrialization, is now in the pilot phase	DOW: Raw material contains part of the crystal water, electrode wear I. Farben: Low current density, short service life
Degree of industrialization	Industrialization [46]	Industrialization [89]	Industrialization [97]	Pilot scale [127,128]	Industrialization [124]
Comprehensive comparison	Good	General	General	Better	General

**Table 9 materials-16-03340-t009:** Comparison of Parameters of Different Reductants.

Reductant	Reducing Reaction	Reaction Condition	Reduction Rate	Energy Consumption t/tce	Total CO_2_ Emissions/t
75% SiFe_(s)_	2(MgO·CaO)_(S)_ + Si_(s)_ = 2Mg_(g)_ + 2CaO·SiO_2(S)_	1473 K	85%	4.549	11.339
75% SiFe_(l)_	2(MgO·CaO)_(s)_ + Si_(l)_ = 2Mg_(g)_+2CaO·SiO_2(s)_	1723 K	95%	4.228	10.541
Al	CaO_(s)_ + 6MgO_(s)_ + 4Al_(l)_ = 6Mg_(g)_ + CaO·2Al_2_O_3(s)_	1473 K	95%	8.735	21.777
C	(MgO·CaO)_(S)_ + C_(s)_ = Mg_(g)_ + CaO_(s)_ + CO_(g)_	1773 K	82%	1.500	3.741
Fe-Si-Al	CaO_(s)_ + 6MgO_(s)_ + 4Al_(l)_ = 6Mg_(g)_ + CaO·2Al_2_O_3(s)_	1473 K	88%	6.419	16.002
	2(MgO·CaO)_(S)_ + Si_(s)_ = 2Mg_(g)_ + 2CaO·SiO_2(s)_	1473 K			
Industrial silicon	2(MgO·CaO)_(S)_ + Si_(s)_ = 2Mg_(g)_ + 2CaO·SiO_2(S)_	1473 K	95%	5.576	13.901

**Table 10 materials-16-03340-t010:** Comparison table showing the parameters and processes in different electrolytic magnesium smelting processes.

Process	Dehydration Equipment	Dehydrate Condition	Stage of Dewatering		
DSM	Fluidized, Chlorinator	Fluidized bed dryer: 403~473 K dehydration 95% Chlorinator: 973~1023 K dehydration 5% [117,119]	Chlorinator first chamber: carnallite melting chamber [117,119]	Chlorinator second room: chlorination reaction chamber [117,119]	Chlorinator third room: sedimentation chamber [117,119]
DOW	Rotary kiln, Spray dryer	Rotary kiln: Calcination of dolomite reacts with magnesium-containing seawater, filtered and reacts with hydrochloric acid Water-containing (27%) magnesium chloride enters the spray dryer [121]	Calcined white + magnesium-containing seawater = calcium hydroxide and magnesium hydroxide [121]	Magnesium hydroxide + hydrochloric acid = magnesium chloride + water; into the dryer MgCl_2_·2H_2_O [121]	Electrolytic cell electrolytic dehydration Electrolytic efficiency 18~19 kWh/kg; graphite consumption 100 g/kg [121]
MagCorp	Spray dryer, Melting tank	Spray dryer: 4% magnesium oxide and 4% water Melting tank: 1088 K anhydrous magnesium chloride [118,120]	Evaporation and concentration of salt lake brine and spray drying to obtain powdered magnesium chloride [118,120]	Magnesium chloride powder placed in a melt tank to obtain purer anhydrous magnesium chloride [118,120]	Preparation of metal magnesium with electrolysis [118,120]
Hydro	Evaporator, Fluidized	Evaporator waste heat heating Fluidized bed heating HCl heating 603 K [46,125]	Preliminary drying: moisture content of 45~50% MgCl_2_·6H_2_O [46,125]	Fluidized bed secondary drying [46,125]	Drying with heated HCl gas [46,125]
Magnola	Spray dryer, Chlorinator	Gaseous HCl Melting temperature 788~888 K Electrolyte temperature 923 K [122,123]	Preparation of magnesium chloride solution with soaking in concentrated hydrochloric acid [122,123]	Spray dryer drying [122,123]	Chlorinator drying [122,123]
AMC	Leaching tank, Separator, Dryer	Chlorine recycling Organic solvent recycling [124,126]	Reaction of magnesite with hydrochloric acid to produce hydrated magnesium chloride [124,126]	Hydrated magnesium chloride forms organic complexes with organic solvents (ethylene glycol or methanol) [124,126]	Separating hexaammine magnesium chloride and heating it to obtain anhydrous magnesium chloride [124,126]

## Data Availability

The relevant data in the article have indicated the source location, and the required data can be obtained in the corresponding references.

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
