# Peer review of "Research on the Process, Energy Consumption and Carbon Emissions of Different Magnesium Refining Processes"

_materials, 2023, doi:10.3390/ma16093340_

Round 1
Reviewer 1 Report
I have some remarks and comments:
1. The abstract repeats what is in the introduction
2. Figure 2 should be of better quality
3. The line 272 what is mean “the thermodynamic temperature”?
4. The description of equation 1 should be clear, it is best to describe its components below the equation
5. Table 2 does not have a defined formula and is well written. I omit the fact that there is no discussion of the results given in Table 2.
6. The description of equation 2 should be clear, it is best to describe its components below the equation
7. Table 3 does not have a defined formula and is well written. I omit the fact that there is no discussion of the results given in Table 3.
In general, I have a problem with this review, because it is a publication that describes and does not bring anything from itself. We have accumulated large amounts of tables, and equations, but there is no discussion of these results. There is no summary at all, and no meaningful conclusion.
Author Response
Manuscript Materials-2297391
- Response to Reviewers
Dear Editor and Reviewers,
Thank you for giving us the opportunity to submit a revised draft of the manuscript “The research on process, energy consumption and carbon emissions of different magnesium refining processes” for publication in the Materials. We appreciate the time and effort that you and the reviewers dedicated to providing feedback on our manuscript and are grateful for the insightful comments on and valuable improvements to our paper.
We have incorporated most of the suggestions made by reviewers. Those changes are highlighted within the manuscript. Please see below, in blue, for a point-by-point response to the reviewers’ comments and concerns. All page numbers refer to the revised manuscript file with tracked changes.
- Comments from the Editors and Reviewers:
- Reviewer1:
- The abstract repeats what is in the introduction.
- Author response:
Thank you for pointing this out. The reviewer is correct. According to your suggestion, we have modified the abstract part, see page 1, lines 7 to 16.
- For ease of reading, we have listed the changes below:
“Under the policy of low carbon energy saving, higher requirements are put forward for magnesium smelting. As the mainstream magnesium smelting process, Pidgeon process has the disadvantages of long production cycle, high energy consumption and high carbon emission, which is difficult to meet the requirements of green environmental protection. This paper reviews the research progress of different magnesium smelting processes, and further analyzes their energy consumption and carbon emissions. It is concluded that the standard coal required for the production of tons of magnesium by the relative vacuum continuous magnesium refining process is reduced by more than 1.5 t, the carbon emission is reduced by more than 10 t, and the reduction cycle is shortened by more than 9.5 h. The process has the advantages of clean, efficient and low carbon, which provides a new way for the development of magnesium industry. ”
- Figure 2 should be of better quality.
- Author response:
Thank the reviewer for the question, the reviewer's point of view is very correct. We redraw Figure 2 and Figure 3, please see page 3, line 84.
- For ease of reading, we have listed the changes below:
|
|
|
|
Before modification(Figure 2) |
Redrawn Figure 2 |
|
|
|
|
Before modification(Figure 3) |
Redrawn Figure 3 |
- The line 272 what is mean “the thermodynamic temperature”?
- Author response:
The reviewer's opinion is very correct. Due to the negligence of the author, the symbols in the formula are not clearly explained. The thermodynamic temperature here refers to the dynamic thermal decomposition temperature, which is used to compare the thermal stability of the material. We have reinterpreted the symbols in the formula, please see page 7, line 223-225.
- For ease of reading, we have listed the changes below:
“Where the α is the conversion rate , Wx is the thermal gravimetric rate at any point in the thermogram, W0 and W∞ are constants, and T is the dynamic thermal decomposition temperature.”
- The description of equation 1 should be clear, it is best to describe its components below the equation.
- Author response:
Thanks for the suggestions put forward by the reviewer, the reviewer is correct. The author has modified the article according to the reviewer's suggestion, please see page 6, lines 217 to page 7, lines 222; page 7, line 223-225.
- For ease of reading, we have listed the changes below:
page 6, lines 217 to page 7, lines 222:
“Liu Xiaoxing et al[59] improved the conversion rate α in the kinetic parameter Freeman Carroll method for the thermal decomposition reaction of solids. The activation energy E of thermal decomposition of Mg(OH)2 was obtained as 122KJ/mol and the number of reaction steps n was 0.68. The results of the improved algorithm were more accurate. The improved calculation formula of conversion rate α is shown in Formula 1.”
Page 7, Line 223-225:
“Where the α is the conversion rate , Wx is the thermal gravimetric rate at any point in the thermogram, W0 and W∞ are constants, and T is the dynamic thermal decomposition temperature.”
- Table 2 does not have a defined formula and is well written. I omit the fact that there is no discussion of the results given in Table 2.
- Author response:
The reviewer's opinion is quite correct. We have supplemented the undefined parts of the table, as detailed on page 7, lines 268 to page 8, lines 272. The results given in table 2 are discussed, please see page 7, lines 261-268.
- For ease of reading, we have listed the changes below:
Page 7, lines 268 to page 8, lines 272:
“In the table, α is the conversion rate, f(α) is the differential form of the kinetic function of the solid phase reaction, g(α) is the integral form of the kinetic function of the solid phase reaction, E is the apparent activation energy, LogA is the pre-exponential constant, n is the dimensionless reaction order, and R2 is the correlation.”
Page 7, lines 261-268:
“Based on the kinetic model analysis, the kinetic function of the solid phase reaction is described according to Table 2 under three assumptions. Firstly, the reaction consists of several basic reaction steps, and the conversion rate of each step can be described by its own kinetic equation. The second assumption is that all kinetic parameters, such as E, A, n ( reaction order ) and f(α), are assumed to be constants for each individual reaction step. The third assumption is that the thermal analysis signal is the sum of the signals of a single reaction step, and the effect of each step is calculated as the conversion rate, multiplied by the effect of the step.”
- The description of equation 2 should be clear, it is best to describe its components below the equation.
- Author response:
Thanks for the suggestions put forward by the reviewer, the reviewer is correct. The author has modified the article according to the reviewer's suggestion, please see page 8, lines 291-297; page 9, lines 298-300.
- For ease of reading, we have listed the changes below:
Page 8, Line 291-297:
“Huaqiang Chu et al.[66] established a single homogeneous chemical reaction model under the condition of silicon thermal reduction and studied the kinetic principle of the reduction process. The equation of the 1273~1473K chemical reaction model is shown in Formula (2). It is concluded that low heat transfer efficiency is an important factor limiting magnesium production capacity in the initial reduction stage. The radiation intensity changes with temperature, and the radiation heat transfer affects the heat transfer process and cannot be ignored or simplified.”
Page 9, Line 298-300:
“Where α is the absorption coefficient, k0 is the pre-constant (min-1), T is the local temperature (K), and τ is the reduction time of magnesium in the experimental and numerical models (min).”
- Table 3 does not have a defined formula and is well written. I omit the fact that there is no discussion of the results given in Table 3.
- Author response:
The reviewer's opinion is quite correct. We have supplemented the undefined parts of the table, as detailed on page 11, lines 382-385. The results given in table 3 are discussed, please see page 11, lines 386-393.
For ease of reading, we have listed the changes below:
Page 11, Line 382-385:
“In the table, ΔG represents the standard Gibbs free energy, TR represents the temperature of the reaction in the standard state, TRP represents the temperature of the reaction under different pressures, X represents the ratio of material to magnesium, and ω represents the carbon emission.”
Page 11, Line 386-393:
“From the thermodynamic point of view, the reaction of aluminothermic reduction of magnesium oxide to 12CaO·7Al2O3, CaO·Al2O3, CaO·2Al2O3, MgO·Al2O3 and Al2O3 can be carried out under vacuum conditions, and the theoretical initial reaction temperature is more than 100K lower than that of silicothermic method. In the case of obtaining the same magnesium reduction rate by Pidgeon method, the reduction temperature of aluminothermic reduction can be reduced by 50℃. The reduction of reaction temperature is beneficial to reduce the energy consumption and prolong the service life of the reduction tank.”
- For editors and reviewers
Once again, we thank you for the time you put in reviewing our paper and look forward to meeting your expectations. Since your inputs have been precious, in the eventuality of the publication, we would like to acknowledge your contribution explicitly.

Reviewer 2 Report
Manuscript number: Materials (ISSN 1996-1944)
Title: The research on process, energy consumption and carbon emis-2 sions of different magnesium refining processes; Authors : Jingzhong Xu 1, Ting-an Zhang 1,* and Xiaolong Li 1
Comments to the Authors
1. This article is well compiled, and review is quite good and exhaustive.
2. The abstract needs revision to bring in novelty of work and the need of such research.
3. In the introduction the authors provided a systematic review of the previous works like what work had been done by the peers in China or across the globe and what processes they have developed/ adopted. Line 44 many researchers …. Provide the references
4. Line 63-67 can be converted or represented in the form of pie chart or graphics.
5. Subsequent paragraph can be rewritten as the primary source and secondary sources of magnesium and the salient properties or characteristics can be presented in tabular form.
6. Authors have provided composition in some places and in some places nothing, keep the same format. E.g., dolomite composition given and magnesite or other not provided.
7. Authors can tabulate the salient properties or characteristics till point 5.
8. Table 9, comparison of various processes – quality or grade of raw material to be provided and the cost unit need to be provided. References for the data is required in the table.
9. Authors may also consider providing information on the scale of all the processes like is it lab scale data or pilot scale or TRL level of these technologies.
10. Title of table 11??? What does it mean?? Where it is referred in the text. Also provide the references.
11. Comparison of parameters of different magnesium smelting methods: table 8 provides information on influencing factors of leaching rate under section 3. Please corelate and place it at appropriate place in the text.
12. Information on techno economics of the best process routes and the scale of the untis if mentioned, this will be a very qualitative information.
13. Are there any process residue or waste generation from any of these processes?? Any information or review will further strengthen the review. Apart from carbon footprint what are the other issues one can foresee.
14. Part 4: too exhaustive, break into paragraphs and provide way forward and to the point conclusions. Any technoeconomic analysis of the process will further help in choosing the processes.
The topic is of good interest and the research covered appears to be comprehensive.
Author Response
Manuscript Materials-2297391
- Response to Reviewers
Dear Editor and Reviewers,
Thank you for giving us the opportunity to submit a revised draft of the manuscript “The research on process, energy consumption and carbon emissions of different magnesium refining processes” for publication in the Materials. We appreciate the time and effort that you and the reviewers dedicated to providing feedback on our manuscript and are grateful for the insightful comments on and valuable improvements to our paper.
We have incorporated most of the suggestions made by reviewers. Those changes are highlighted within the manuscript. Please see below, in blue, for a point-by-point response to the reviewers’ comments and concerns. All page numbers refer to the revised manuscript file with tracked changes.
- Comments from the Editors and Reviewers:
- Reviewer2:
- This article is well compiled, and review is quite good and exhaustive.
- Author response:
Thank you very much for the reviewer's affirmation of our manuscript, and also for the reviewer's energy in reviewing the manuscript.
- The abstract needs revision to bring in novelty of work and the need of such research.
- Author response:
Thank you for pointing this out. The reviewer is correct. According to your suggestion, we have modified the abstract part, see page 1, lines 7 to 16.
- For ease of reading, we have listed the changes below:
“Under the policy of low carbon energy saving, higher requirements are put forward for magnesium smelting. As the mainstream magnesium smelting process, Pidgeon process has the disadvantages of long production cycle, high energy consumption and high carbon emission, which is difficult to meet the requirements of green environmental protection. This paper reviews the research progress of different magnesium smelting processes, and further analyzes their energy consumption and carbon emissions. It is concluded that the standard coal required for the production of tons of magnesium by the relative vacuum continuous magnesium refining process is reduced by more than 1.5 t, the carbon emission is reduced by more than 10 t, and the reduction cycle is shortened by more than 9.5 h. The process has the advantages of clean, efficient and low carbon, which provides a new way for the development of magnesium industry. ”
- In the introduction the authors provided a systematic review of the previous works like what work had been done by the peers in China or across the globe and what processes they have developed/ adopted. Line 44 many researchers Provide the references.
- Author response:
Thank you very much for the reviewer's affirmation of the introduction, and the reviewer's suggestions are very correct. We have annotated the references to 'many researchers...', please see page 2, lines 46-48.
- For ease of reading, we have listed the changes below:
“Many researchers have studied the influence on the dynamic performance[46.77.99], heat transfer performance[52.80.107] and processing parameters[66.85.120] of magnesium smelting process.”
- Line 63-67 can be converted or represented in the form of pie chart or graphics.
- Author response:
Thank you for pointing this out. The reviewer 's suggestion is very correct. We modify lines 63-67 of the manuscript and present some information in the form of a pie chart, as shown on page 2, lines 63-70. Add pie chart 2.
- For ease of reading, we have listed the changes below:
Page 2, lines 63-70
“According to the statistics of the Magnesium Industry Branch of the Non-ferrous Metals Association[11], the demand for raw magnesium is concentrated in the metallurgical field ( aluminum alloy addition, steelmaking desulfurization, ball milling cast iron and metal reduction ) and processing field (rare earth magnesium alloy, castings, die castings and profiles). The domestic original magnesium capacity concentration is low, more than 30,000 t of production capacity of only 13 companies, a total market share of 35.9%.The distribution of domestic magnesium smelters and raw magnesium production in 2020 is shown in Figure 2[12-19].”
Pie chart 2:
Figure 2. Distribution of magnesium smelter and raw magnesium production in 2020
- Subsequent paragraph can be rewritten as the primary source and secondary sources of magnesium and the salient properties or characteristics can be presented in tabular form.
- Author response:
Thank you for pointing this out. The reviewer's opinion is very correct. According to this suggestion, readers can have a more intuitive understanding. On the advice of the reviewer, the author has rearranged page 3, lines 80 to page 4, lines 148 of the manuscript and converted the main properties into tabular form. Please see page 3, lines 73-82. The main properties of magnesium raw materials are shown in Table 1. Please see page 3, lines 83.
- For ease of reading, we have listed the changes below:
Page 3, lines 73-82:
“The magnesium refining methods are divided into thermal reduction and electrolysis, and the sources of raw materials are magnesite, dolomite, hydromagnesite, halloysite, serpentine and seawater[15-19]. Different raw materials differ in magnesium content, production methods and sources, and the main source methods are mining, open pit mining, seawater and salt lake reprocessing, asbestos production waste, etc. Magnesium is obtained from different magnesium sources, usually as natural raw materials, and rarely in pure form. The main properties of different raw materials are shown in Table 1. The raw materials for magnesium smelting are mainly magnesite and serpentine. The distribution of magnesite production is shown in Figure 3, and the distribution of serpentine production is shown in Figure 4.”
Page 3, lines 83
|
Raw material |
Essential component |
Ore color |
Crystalline phase |
Hardness |
Density (kg/m3) |
Reserve (million tons) |
|
Magnesite |
MgCO3[10] |
White[10] |
Tripartite crystal system[20] |
4-4.5[20.21] |
2.9-3.1[22] |
40[23] |
|
Dolomite |
CaMg(CO3)2[24] |
Off-white |
Tripartite crystal system[25.26] |
3.5-4[27] |
2.8-2.9[28] |
85[29] |
|
Bichofite |
MgCl2·6H2O[30] |
White |
Monoclinic system[31] |
1-2 |
—— |
17.74 % of salt lake brine[32] |
|
Carnallite |
KCl·MgCl2·6H2O |
White、Red |
Hexahedron[33] |
—— |
1.6 |
36.8 % of the Dead Sea[34.35] |
|
Serpentine |
Mg6[Si4O10](OH)8 |
Waxy luster |
Monoclinic system[36-38] |
2.5-4[39] |
2.57[40] |
120[41] |
|
Seawater |
Table2[42-44] |
Dark blue |
—— |
—— |
1.02-1.07[45] |
—— |
Table 1. Main properties of raw materials for magnesium smelting
- Authors have provided composition in some places and in some places nothing, keep the same format. E.g., dolomite composition given and magnesite or other not provided.
- Author response:
Thank you very much for the reviewer' s advice, the reviewer is correct. According to the reviewer's suggestion, the main components of all magnesium raw materials are summarized in table 1. Please see page 3, lines 83.
- For ease of reading, we have listed the changes below:
The complete table is given in question 5, and only the main components are given here.
|
Raw material |
Essential component |
... |
|
Magnesite |
MgCO3[10] |
... |
|
Dolomite |
CaMg(CO3)2[24] |
... |
|
Bichofite |
MgCl2·6H2O[30] |
... |
|
Carnallite |
KCl·MgCl2·6H2O |
... |
|
Serpentine |
Mg6[Si4O10](OH)8 |
... |
|
Seawater |
Table2[42-44] |
... |
- Authors can tabulate the salient properties or characteristics till point 5.
- Author response:
Thank you for pointing this out. The author has listed the prominent nature or characteristics in Table 1 according to the opinions of the reviewers. Please see page 3, lines 83.
- For ease of reading, we have listed the changes below:
|
Raw material |
Essential component |
Ore color |
Crystalline phase |
Hardness |
Density (kg/m3) |
Reserve (million tons) |
|
... |
... |
... |
... |
... |
... |
... |
- Table 9, comparison of various processes-quality or grade of raw material to be provided and the cost unit need to be provided. References for the data is required in the table.
- Author response:
Thank you for pointing this out. The author has added various process raw material quality in Table 9 and provided cost units. The data involved in the table provide references. Please see page 35, lines 1296.
- For ease of reading, we have listed the changes below:
|
Comparative content |
Silicothermic |
Aluminothermic |
Carbothermic |
Relative vacuum |
Electrolytic method |
|
... |
... |
... |
... |
... |
... |
|
Quality of raw material (MgO%) |
23.8%(Dolomite) 46.0%(Magnesite) |
23.8%(Dolomite) 46.0%(Magnesite) |
23.8%(Dolomite) 46.0%(Magnesite) |
23.8%(Dolomite) 46.0%(Magnesite) |
—— |
|
... |
... |
... |
... |
... |
... |
|
Cost (Production cost per t/¥) |
20970[51] |
21567(3CaO·Al2O3) 19015(CaO·2Al2O3) 19631(CaO·Al2O3) 26380(12CaO·7Al2O3) [78.79.82.84.82] |
11980[104.106] |
20843(3CaO·Al2O3) 18318(CaO·2Al2O3) 18780(CaO·Al2O3) 25445(12CaO·7Al2O3) [133.134] |
16520[123.126] |
|
... |
... |
... |
... |
... |
... |
- Authors may also consider providing information on the scale of all the processes like is it lab scale data or pilot scale or TRL level of these technologies.
- Author response:
Thank you very much for the reviewer 's suggestion, the author has listed the degree of industrialization of different processes in Table 9. Please see page 35, line 1296.
- For ease of reading, we have listed the changes below:
|
Comparative content |
Silicothermic |
Aluminothermic |
Carbothermic |
Relative vacuum |
Electrolytic method |
|
... |
... |
... |
... |
... |
... |
|
Degree of industrialization |
Industrialization[49] |
Industrialization[92] |
Industrialization[101] |
Pilot scale[133.134] |
Industrialization[130] |
|
... |
... |
... |
... |
... |
... |
- Title of table 11? What does it mean? Where it is referred in the text. Also provide the references.
- Author response:
Thank you for pointing this out. Thanks to the reviewer's suggestion, the author has revised the name of Table 11. Table 11 refers to page 37, lines 1335-1339. And complete the references involved in the table. Please see page 37-38, line 1339.
- For ease of reading, we have listed the changes below:
Table 11. Comparison table of parameters and processes of different electrolytic magnesium smelting processes
|
Process |
Dehydration equipment |
Dehydrate condition |
Stage of dewatering |
||
|
DSM |
Fluidized, Chlorinator |
Fluidized bed dryer: 403K~473K dehydration 95% Chlorinator: 973K~1023K dehydration 5%[122.124] |
Chlorinator first chamber: carnallite melting chamber[122.124] |
Chlorinator second room: chlorination reaction chamber[122.124] |
Chlorinator third room: sedimentation chamber[122.124] |
|
DOW |
Rotary kiln, Spray dryer |
Rotary kiln: Calcination of dolomite reacts with magnesium-containing seawater, filtered and reacts with hydrochloric acid Water-containing (27%) magnesium chloride enters the spray dryer[126] |
Calcined white+magnesium-containing seawater=calcium hydroxide and magnesium hydroxide[126] |
Magnesium hydroxide+hydrochloric acid=magnesium chloride+water; into the dryer MgCl2·2H2O[126] |
Electrolytic cell electrolytic dehydration Electrolytic efficiency 18~19kWh/kg; graphite consumption 100g/kg[126] |
|
MagCorp |
Spray dryer, Melting tank |
Spray dryer: 4% magnesium oxide and 4% water Melting tank: 1088K anhydrous magnesium chloride [123.125] |
Evaporation and concentration of salt lake brine and spray drying to obtain powdered magnesium chloride[123.125] |
Magnesium chloride powder placed in a melt tank to obtain purer anhydrous magnesium chloride[123.125] |
Preparation of metal magnesium by electrolysis[123.125] |
|
Hydro |
Evaporator, Fluidized |
Evaporator waste heat heating Fluidized bed heating HCl Heating 603K[129.131] |
Preliminary drying: moisture content of 45%~50% MgCl2·6H2O[129.131] |
Fluidized bed secondary drying[129.131] |
Drying with heated HCl gas[129.131] |
|
Magnola |
Spray dryer, Chlorinator |
Gaseous HCl Melting temperature 788K~888K Electrolyte temperature 923K[127.128] |
Preparation of Magnesium Chloride Solution by Soaking in Concentrated Hydrochloric Acid[127.128] |
Spray dryer drying[127.128] |
Chlorinator drying[127.128] |
|
AMC |
Leaching tank, Separator, Dryer |
Chlorine recycling Organic solvent recycling[130.132] |
Reaction of Magnesite with Hydrochloric Acid to Produce Hydrated Magnesium Chloride[130.132] |
Hydrated magnesium chloride forms organic complexes with organic solvents (ethylene glycol or methanol)[130.132] |
Separating hexaammine magnesium chloride and heating it to obtain anhydrous magnesium chloride[130.132] |
- Comparison of parameters of different magnesium smelting methods: table 8 provides information on influencing factors of leaching rate under section 3. Please corelate and place it at appropriate place in the text.
- Author response:
Thank you for pointing this out. Thank you very much for the reviewer' s suggestion. The author has revised the position of table 8. And modify the text information corresponding to the tables in the manuscript. Please see page 33, line 1206-1211; page 33, line 1212; page 33, line 1213-1221.
- For ease of reading, we have listed the changes below:
Page 33, line 1206-1211:
“Yanwen Yan[135] et al.proposed the 'Ammonia leaching calcium-carbonation calcium fixation' process to treat a large amount of calcium-containing solid waste produced by silicothermic reduction of magnesium smelting. Using NH4Cl solution as leaching agent, the effects of leaching temperature, leaching time, liquid-solid ratio, NH4Cl mass fraction and magnesium slag particle size on the leaching effect of calcium in slag were analyzed. The data are shown in Table 8.”
Page 33, line 1212:
Table 8. Table of influencing factors of leaching rate[135]
|
Influencing factors |
Leaching temperature(℃) |
Leaching time(h) |
Liquid-solid ratio |
NH4Cl mass fraction(%) |
Magnesium slag particle size(μm) |
|
At different levels ( leaching rate ) |
60(33%) |
1(37%) |
6(34%) |
26(43%) |
150~270(29%) |
|
70(36%) |
2(55%) |
8(46%) |
28(48%) |
96~150(43%) |
|
|
80(44%) |
3(71%) |
10(56%) |
30(54%) |
74~96(52%) |
|
|
90(56%) |
4(77%) |
12(57%) |
32(56%) |
<74(56%) |
Page 33, line 1213-1221:
“It is analyzed that when the temperature is in the range of 328~363K, the mass fraction of NH4Cl is 30%, and the calcium leaching rate is 80%:p(NH3)/pθ=0.2, the Gibbs free energy ΔG of Ca2SiO4 phase leaching reaction in magnesium slag is less than 0, and Ca2SiO4 can be decomposed into Ca2+ solution. The results show that with the increase of reaction temperature, the finer the particle size of magnesium slag, the increase of liquid-solid ratio and the increase of NH4Cl mass fraction, the leaching effect of calcium will be enhanced. The leaching rate of calcium in magnesium slag can reach 77.01% under the conditions of NH4Cl mass fraction of 30%, leaching temperature of 90℃, liquid-solid ratio of 10 and leaching time of 4h.”
- Information on techno economics of the best process routes and the scale of the untis if mentioned, this will be a very qualitative information.
- Author response:
Thank you for pointing this out. The author modifies the conclusion part and supplements the information of the best process route and scale of technical economy. Please see page 39, line 1341-1361.
- For ease of reading, we have listed the changes below:
“Compared with the Pidgeon process, the consumption of standard coal per ton of magnesium in the vacuum continuous magnesium smelting process is 3 ~ 3.5t, which is reduced by more than 33.5 %. The reduction cycle is shortened from 10-14h to 1-1.5h, and the carbon emission per ton of magnesium is reduced from 23t to 11-13t. Most importantly, breaking the limitation of vacuum conditions, continuous production can be achieved, and magnesium production can be increased by more than 55 %. The CO2 in the calcination process is absorbed by calcium-containing alkaline waste slag, and the reduction slag can be used as refractory material to achieve zero solid waste, low energy consumption and low carbon emission. It provides a new way for green metallurgy.”
- Are there any process residue or waste generation from any of these processes?? Any information or review will further strengthen the review. Apart from carbon footprint what are the other issues one can foresee.
- Author response:
Thank you for pointing this out. The author supplements the problems existing in different processes (Please see page 39, line 1348-1352)and adds the characteristics of zero solid waste relative to the relative vacuum continuous magnesium smelting process (Please see page 39, line 1358-1361).
- For ease of reading, we have listed the changes below:
Page 39, line 1348-1352:
“The high energy consumption of molten salt electrolysis limits its application range. In the thermal reduction method, the aluminothermic reducing agent is expensive ; carbon thermal method pollutes the environment seriously ; although the silicon thermal method is widely used in industry, it is limited by vacuum conditions and cannot be continuously produced.”
Page 39, line 1358-1361:
“The CO2 in the calcination process is absorbed by calcium-containing alkaline waste slag, and the reduction slag can be used as refractory material to achieve zero solid waste, low energy consumption and low carbon emission. It provides a new way for green metallurgy.”
- Part 4: too exhaustive, break into paragraphs and provide way forward and to the point conclusions. Any technoeconomic analysis of the process will further help in choosing the processes.
- Author response:
Thank you for pointing this out. The reviewer's suggestion is very correct, and the author simplifies the conclusion. Please see page 39, line 1341-1361.
- For ease of reading, we have listed the changes below:
“Under the policy of energy conservation, emission reduction and environmental protection, the comprehensive energy consumption per ton of magnesium in magnesium smelting enterprises is controlled below 4.5 t standard coal. The consumption of dolomite is less than 10.5 t, the consumption of ferrosilicon ( Si > 75 % ) is less than 1.05 t, and the consumption of fresh water is less than 10 t. The recovery rate of magnesium is higher than 80 %, the utilization rate of silicon is higher than 75 %, and the recovery rate of crude magnesium refining is higher than 96 %. The comprehensive utilization rate of reduction slag is higher than 70 %. The high energy consumption of molten salt electrolysis limits its application range. In the thermal reduction method, the aluminothermic reducing agent is expensive ; carbon thermal method pollutes the environment seriously ; although the silicon thermal method is widely used in industry, it is limited by vacuum conditions and cannot be continuously produced.
Compared with the Pidgeon process, the consumption of standard coal per ton of magnesium in the vacuum continuous magnesium smelting process is 3 ~ 3.5t, which is reduced by more than 33.5 %. The reduction cycle is shortened from 10-14h to 1-1.5h, and the carbon emission per ton of magnesium is reduced from 23t to 11-13t. Most importantly, breaking the limitation of vacuum conditions, continuous production can be achieved, and magnesium production can be increased by more than 55 %. The CO2 in the calcination process is absorbed by calcium-containing alkaline waste slag, and the reduction slag can be used as refractory material to achieve zero solid waste, low energy consumption and low carbon emission. It provides a new way for green metallurgy.”
- For editors and reviewers
Once again, we thank you for the time you put in reviewing our paper and look forward to meeting your expectations. Since your inputs have been precious, in the eventuality of the publication, we would like to acknowledge your contribution explicitly.

Round 2
Reviewer 1 Report
The publication is now more readable and transparent in relation to the previous version. Particularly noteworthy are the beautiful drawings and diagrams, also the abstract does not repeat the introduction and we have an extensive conclusion. I would ask for careful editing of style and language.